# 🛸 UFO-4D: Unposed Feedforward 4D Reconstruction from Two Images

**Junhwa Hur**[1]  **Charles Herrmann**[1]  **Songyou Peng**[1]  **Philipp Henzler**[1]  **Zeyu Ma**[2]
**Todd Zickler**[3]  **Deqing Sun**[1]

[1]Google  [2]Princeton University  [3]Harvard University

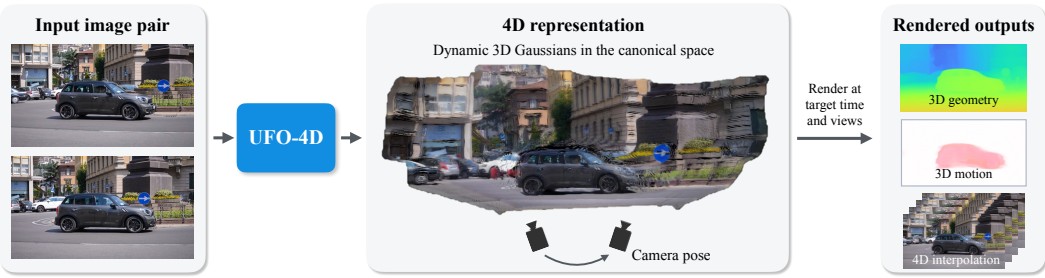

Figure 1: Given a pair of unposed images, the proposed UFO-4D outputs dynamic 3D Gaussians in the canonical space and relative camera pose in a feedforward manner. This explicit 4D representation can solve various downstream tasks such as 3D geometry (point, depth) and motion (scene flow, optical flow). Besides, it can interpolate image, geometry, and motion at novel view and time.

## Abstract

Dense 4D reconstruction from unposed images remains a critical challenge, with current methods relying on slow test-time optimization or fragmented, task-specific feedforward models. We introduce UFO-4D, a unified feedforward framework to reconstruct a dense, explicit 4D representation from just a pair of unposed images. UFO-4D directly estimates dynamic 3D Gaussian Splats, enabling the joint and consistent estimation of 3D geometry, 3D motion, and camera pose in a feedforward manner. Our core insight is that differentiably rendering multiple signals from a single Dynamic 3D Gaussian representation offers major training advantages. This approach enables a self-supervised image synthesis loss while tightly coupling appearance, depth, and motion. Since all modalities share the same geometric primitives, supervising one inherently regularizes and improves the others. This synergy overcomes data scarcity, allowing UFO-4D to outperform prior work by up to $3\times$ in joint geometry, motion, and camera pose estimation. Our representation also enables high-fidelity 4D interpolation across novel views and time. Please visit our project page for visual results: https://ufo-4d.github.io/.

## 1 Introduction

Joint estimation of camera pose, 3D geometry, and 3D motion – a task known as 4D scene reconstruction – from casually captured images is a fundamental challenge in computer vision with applications in robotics (Xu et al., 2024), autonomous driving (Geiger et al., 2012), and 3D/4D generative AI (Liang et al., 2024; Xie et al., 2025; Yu et al., 2024). Recovering dense 4D information from limited 2D inputs, however, is an inherently ill-posed problem. While data-driven approach can be a solution, a challenge continues with a scarcity of dense, large-scale 4D training data. Synthetic datasets (Zheng et al., 2023) provide dense, pixel-perfect supervision, yet they often suffer from a significant domain gap and a lack of diversity. Real-world data (Jin et al., 2025) is generally constrained to sparse and noisy annotations, limiting its effectiveness for training robust models.

Consequently, dynamic scene reconstruction has traditionally centered on slow, test-time optimization pipelines This approach relies on intermediate 2D signals such as depth and optical flow, which imposes a high computational cost and caps performance at the quality of these inputs. A recent shift towards feedforward models, including DUST3R (Wang et al., 2024b), MonST3R (Zhang et al., 2025a), and DynaDUSt3R (Jin et al., 2025), has yielded impressive results on individual perception tasks; however, a single, unified feedforward architecture capable of holistically addressing the full spectrum of 2D and 3D perception tasks does not yet exist. This lack of a unified representation prevents models from exploiting the tightly coupled nature of geometry and motion.

To address these limitations, we introduce UFO-4D, a unified feedforward model for dense 4D reconstruction from just two unposed images. UFO-4D predicts an explicit and dense spatio-temporal representation, Dynamic 3D Gaussian Splatting (D-3DGS), in a single forward pass. This unified representation enables the consistent, integrated estimation of multiple 2D and 3D reconstruction tasks, including depth, scene flow, and optical flow, all from the same underlying model.

Furthermore, our unified representation unlocks two key advantages during training. First, differentiable rendering provides a dense, self-supervised photometric loss that is independent of sparse or noisy ground-truth labels. Second, because 3D geometry and motion originate from the same set of Gaussian primitives, their respective losses are tightly coupled in 3D. This coupling creates a regularizing effect where each supervisory signal helps compensate for imperfections in the other. As a result, UFO-4D achieves state-of-the-art performance on geometry and motion benchmarks, achieving more than $3\times$ lower EPE on Stereo4D and KITTI than competing methods. Furthermore, our explicit 4D representation enables novel applications, including high-fidelity spatio-temporal interpolation of appearance, geometry, and motion from a single prediction.

Our primary contributions are:

- A unified model for unposed feedforward 4D reconstruction from two images using a dynamic 3D Gaussian Splatting representation.
- A robust semi-supervision framework that leverages differentiably rendered outputs to mitigate the scarcity of annotated data.
- 4D spatio-temporal interpolation of image, depth, and motion as a new application of the feedforward output.
- State-of-the-art performance on 3D geometry and 3D motion benchmark datasets.

## 2 RELATED WORK

**3D reconstruction of static scenes.** The field of 3D reconstruction of static scenes, such as multiview geometry (Bae et al., 2022; Ding et al., 2022; Zhang et al., 2023), structure from motion (SfM) (Lindenberger et al., 2021; Pan et al., 2024; Wang et al., 2024a), and simultaneous localization and mapping (SLAM) (Lipson et al., 2024; Teed & Deng, 2021; Zhu et al., 2024), has been also significantly advanced by learning-based approaches. Canonical approaches (Agarwal et al., 2011; Furukawa et al., 2010; Schonberger & Frahm, 2016) rely on a well-established pipeline based on projective geometry, such as feature extraction, triangulation, bundle adjustment, *etc*. However, learning-based approaches demonstrate that each component in the pipeline can be advanced by data-driven approaches. Even the whole pipeline can be end-to-end (Elflein et al., 2025; Liang et al., 2025a; Wang & Agapito, 2025; Wang et al., 2025a; 2024b; Weinzaepfel et al., 2023; Zhang et al., 2025b) when networks learn strong priors from large-scale data. However, these methods specifically target static scenes that satisfy the epipolar geometry constraint.

**3D reconstruction of dynamic scenes.** Traditional approaches (Kumar et al., 2017; Luiten et al., 2020; Mustafa et al., 2016) often tackle this problem using multi-stage pipelines that combine given depth and motion cues and optimize the main objective using rigidity or piecewise smoothness priors about geometry and motion. Recent approaches demonstrate that this highly ill-posed problem can simply be solved by learning strong priors on geometry of dynamic scenes. Those strong prior can be obtained by light-weight finetuning (Lu et al., 2025; Zhang et al., 2025a) of static 3D reconstruction model (Wang et al., 2024b), large-scale training (Wang et al., 2025c), or repurposing a generative model (Mai et al., 2025). While these feedforward methods generate per-frame pointclouds, they lack temporal correspondence, limiting their application to motion understanding tasks.

**Dense 4D reconstruction.** Dense 4D reconstruction methods mainly fall into two categories. One line of work relies on per-scene test-time optimization (Lei et al., 2025; Liu et al., 2025; Wang et al., 2023; 2025b). The approaches show high-fidelity reconstruction, but often take hours to run and depend on pre-computed cues such as camera poses and optical flow. To enable real-time application, other feedforward approaches have been proposed (Feng et al., 2025; Han et al., 2025; Liang et al., 2025b; Sucar et al., 2025), represented in sparse pointmaps. Concurrent works are primarily designed for novel view synthesis and require camera poses at test time (Lin et al., 2026; 2025; Xu et al., 2025b), while others require training separate heads for each downstream task (Badki et al., 2026). In contrast, our method reconstructs a dense and explicit 4D representation in a feedforward manner without requiring camera poses as input. Moreover, our holistic approach is suitable for both reconstruction and synthesis, demonstrating that synthesis improves reconstruction tasks.

**Dense 4D datasets.** The lack of large-scale, densely annotated 4D datasets is a major bottleneck for the development of dense 4D reconstruction. While synthetic data (Zheng et al., 2023) suffers from a domain gap, the recent real-world Stereo4D dataset (Jin et al., 2025) provides valuable supervision, though its annotations remain sparse and can be unreliable in occluded or distant regions. UFO-4D, which uses a unified dynamic 3D Gaussian representation, effectively leverages self-supervision to overcome the limitations of sparse ground truth annotations in existing datasets.

## 3 FEEDFORWARD 4D RECONSTRUCTION FROM TWO UNPOSED IMAGES

Our method, UFO-4D, performs feedforward 4D reconstruction from a pair of unposed images. As in Fig. 1, given two images and their camera intrinsics, UFO-4D directly predicts the relative camera pose and a set of dynamic 3D Gaussians in the coordinate system of the first camera. This explicit 4D representation enables not only the synthesis of novel views and interpolated time steps, but also the direct rendering of various geometric and motion attributes, *e. g.*, 3D point and 3D scene flow. As a result, we can leverage both view synthesis and downstream reconstruction tasks for supervision.

**Formulation.** Given input unposed images $\mathbf{I}_t$ and $\mathbf{I}_{t+1}$, our model $f_\theta$ maps them to the scene representation $(\mathcal{G}, \mathbf{P})$, consisting of a set of dynamic 3D Gaussians $\mathcal{G}$ and a relative camera pose $\mathbf{P}$:

$$f_\theta(\mathbf{I}_t, \mathbf{I}_{t+1}) \mapsto (\mathcal{G}, \mathbf{P}), \quad \text{with} \ \ \mathcal{G} = \{(\boldsymbol{\mu}, \mathbf{v}, \mathbf{r}, \mathbf{s}, \mathbf{h}, o)_{\mathbf{p}} \,|\, \mathbf{p} \in \mathcal{D}(\mathbf{I}_t) \cup \mathcal{D}(\mathbf{I}_{t+1})\}, \quad (1)$$

where the set $\mathcal{D}(\mathbf{I})$ denotes a set of pixels in the image $\mathbf{I}$. Each dynamic 3D Gaussian (Luiten et al., 2024) comprises its 3D center $\boldsymbol{\mu} \in \mathbb{R}^3$, 3D motion $\mathbf{v} \in \mathbb{R}^3$, covariance matrix parameterized by quaternion rotation $\mathbf{r} \in \mathbb{R}^4$ and size $\mathbf{s} \in \mathbb{R}^3$, view-dependent color represented by spherical harmonics in each color channel $\mathbf{h} \in \mathbb{R}^k$, and opacity $o \in [0, 1]$. Our model estimates one 3D Gaussian for each pixel from each image $\mathbf{I}_t$ and $\mathbf{I}_{t+1}$, with the 3D motion $\mathbf{v}$ for Gaussians of $\mathbf{I}_t$ being forward motion ($t{\to}t+1$) and those for $\mathbf{I}_{t+1}$ being backward motion ($t+1{\to}t$). Before rasterization, we translate the 3D center of Gaussians for $\mathbf{I}_{t+1}$ with their motion (*i. e.* $\boldsymbol{\mu}+\mathbf{v}$) and inverse their motion vector (*i. e.* $-\mathbf{v}$) to align with the Gaussians for $\mathbf{I}_t$ to be at the time step. The union of the Gaussians from the two images explicitly represents the dense 4D scene elements that are visible in one or both images. All of the Gaussians $\mathcal{G}$ and the relative camera pose $\mathbf{P}$ are defined in the coordinate of first image $\mathbf{I}_t$, which serves as the canonical space. Following (Ye et al., 2025), we additionally input camera intrinsics, which are commonly available in commercial devices.

**Network architecture.** Figure 2 presents an overview of our network architecture, inspired by DUSt3R (Wang et al., 2024b) and NoPoSplat (Ye et al., 2025). The weight-sharing encoder (Dosovitskiy et al., 2021) processes each input image into image tokens separately. The encoded tokens are concatenated with an intrinsic token and a pose token and then fed into a ViT-based decoder. The intrinsic token is obtained after feeding camera intrinsics into a linear layer (Ye et al., 2025). Pose token is a learnable parameter that is optimized during training. Here, cross-attention layers within each attention block are responsible for matching and integrating information between the two input images (An et al., 2025; Chen et al., 2025).

Next, we connect heads to read out parameters of dynamic 3D Gaussians for each image and a relative camera pose as defined in Eq. (1). The center head predicts the 3D center $\boldsymbol{\mu}$ of each Gaussian. The attributes head predicts its quaternion rotation $\mathbf{r}$, scale $\mathbf{s}$, spherical harmonics $\mathbf{h}$ and opacity $o$; and a velocity head predicts its 3D motion vector $\mathbf{v}$. These heads use a DPT-based architecture (Ranftl et al., 2021) with different output channel dimensions. The pose head, a 3-layer MLP, predicts

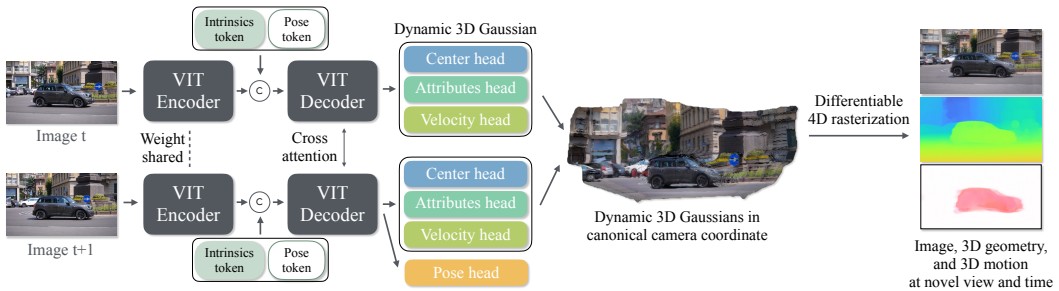

Figure 2: **Network architecture.** Given a pair of input images $\mathbf{I}_t$ and $\mathbf{I}_{t+1}$ and camera intrinsics $\mathbf{K}$, UFO-4D outputs parameters for dynamic 3D Gaussians and relative camera pose in a feed-forward manner. Given the estimates, UFO-4D can render image, point, and motion at any interpolated time and view. While the intrinsic token needs the real camera intrinsics, the pose token is a learnable parameter and does not require the inference-time pose input.

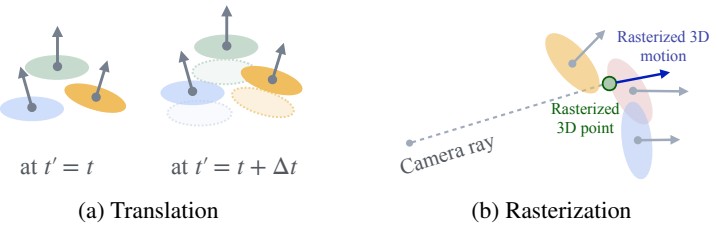

|  (a) Translation | (b) Rasterization |

Figure 3: *(a)* Each Gaussian is translated with its motion to represent 3D scene at time $t + \Delta t$. *(b)* Point and motion as well as an image are rasterized together.

the relative camera pose $\mathbf{P}$ comprising a translation $\boldsymbol{\tau}$ and quaternion $\mathbf{q}$. The estimated pose is used for rasterizing image, depth, and motion at time $t + 1$ during both training and inference time. This direct estimation of pose not only makes inference-time post regression (Wang et al., 2024b; Zhang et al., 2025a) unnecessary but also gives better accuracy (Table G in Appendix).

**Differentiable 4D rasterization.** Our method relies on a unified and differentiable rasterization process that is critical for both model training and downstream 4D perception tasks. We extend the standard 3D Gaussian rasterizer to render not only color images but also dense pointmaps and scene flow at any intermediate time $t' \in [t, t+1]$. To achieve this, we first model the scene at a continuous time $t' = t + \Delta t$ (where $\Delta t \in [0, 1]$) by assuming linear motion. The set of Gaussians $\mathcal{G}(t')$ is formed by updating the position of each predicted Gaussian along its velocity vector:

$$\mathcal{G}(t') = \{(\boldsymbol{\mu} + \Delta t \cdot \mathbf{v}, \mathbf{v}, \mathbf{r}, \mathbf{s}, \mathbf{h}, \mathbf{c}, o)_\mathbf{p}\}, \tag{2}$$

with $\mathbf{p} \in \mathcal{D}(\mathbf{I}_t) \cup \mathcal{D}(\mathbf{I}_{t+1})$.

The temporal evolution is shown in Fig. 3a. To render an image, we then follow the standard alpha-blending pipeline from 3DGS (Kerbl et al., 2023; Luiten et al., 2024). For a given camera view, the color $\hat{\mathbf{I}}_{t'}(\mathbf{p})$ at each pixel is computed by $\alpha$-blending of $\mathcal{N}$-ordered depth-sorted Gaussians along each ray:

$$\hat{\mathbf{I}}_{t'}(\mathbf{p}) = \sum_{i \in \mathcal{N}_\mathbf{p}^{t'}} \mathbf{c}_i o_i \prod_{j=1}^{i-1} (1 - o_j), \tag{3}$$

where the set $\mathcal{N}_\mathbf{p}^{t'}$ contains all the Gaussians that contribute to pixel $\mathbf{p}$, $\mathbf{c}_i$ is the view-dependent color derived from spherical harmonics, and $o$ is the opacity computed from the projected 2D Gaussian. Note that, depth orders of each Gaussian can change with $\Delta t$, and this rasterization process naturally handles occlusion and disocclusion.

A key aspect of our method is applying this rendering principle to geometry and motion. While prior work often uses only the sparse Gaussian primitive centers (Luiten et al., 2024), we produce

dense, geometrically consistent maps by substituting the color $\mathbf{c}_i$ in the blending formula with other intrinsic Gaussian attributes. Toward more precise and dense reconstruction of 3D surface and motion, we render points $\mathbf{X}_{t'} \in \mathbb{R}^3$ and 3D scene flow maps $\mathbf{V}_{t'} \in \mathbb{R}^3$ as follows:

$$\mathbf{X}_{t'}(\mathbf{p}) = \sum_{i \in \mathcal{N}_{\mathbf{p}}^{t'}} \boldsymbol{\mu}_i o_i \prod_{j=1}^{i-1} (1 - o_j), \tag{4a}$$

$$\mathbf{V}_{t'}(\mathbf{p}) = \sum_{i \in \mathcal{N}_{\mathbf{p}}^{t'}} \mathbf{v}_i o_i \prod_{j=1}^{i-1} (1 - o_j). \tag{4b}$$

This unified approach, visualized in Fig. 3b, is fully differentiable (Matsuki et al., 2024). The critical advantage is that it allows supervision signals from rendered images, point maps, and flow maps to be backpropagated through the rasterizer, enabling robust, joint optimization of all heads with a single framework.

**Loss.** Our method takes a semi-supervised learning approach. The total loss $L_{\text{total}}$ consists of a supervised loss $L_{\text{sup}}$ and a self-supervised loss $L_{\text{self}}$:

$$L_{\text{total}} = L_{\text{sup}} + L_{\text{self}}. \tag{5}$$

The supervised loss $L_{\text{sup}}$ penalizes the differences between predictions and ground truth for motion, point, and pose:

$$L_{\text{sup}} = L_{\text{motion}} + w_{\text{point}} L_{\text{point}} + w_{\text{pose}} L_{\text{pose}}, \tag{6a}$$

$$L_{\text{motion}} = \sum_{u \in \{t, t+1\}} \frac{1}{|\mathbf{V}_u^{\text{GT}}|} \left( ||\mathbf{v}_u - \mathbf{V}_u^{\text{GT}}|| + ||\mathbf{V}_u - \mathbf{V}_u^{\text{GT}}|| \right), \tag{6b}$$

$$L_{\text{point}} = \sum_{u \in \{t, t+1\}} \frac{1}{|\mathbf{X}_u^{\text{GT}}|} \left( ||\boldsymbol{\mu}_u - \mathbf{X}_u^{\text{GT}}|| + ||\mathbf{X}_u - \mathbf{X}_u^{\text{GT}}|| \right), \tag{6c}$$

$$L_{\text{pose}} = ||\mathbf{q} - \mathbf{q}_{\text{GT}}|| + ||\boldsymbol{\tau} - \boldsymbol{\tau}_{\text{GT}}||. \tag{6d}$$

We apply $L_{\text{motion}}$ at each Gaussian center's motion $\mathbf{v}$ as well as rasterized motion $\mathbf{V}$, normalized by the number of valid ground truth $|\mathbf{V}_u^{\text{GT}}|$, and the same for $L_{\text{point}}$, We enforce $L_{\text{pose}}$ for translation and quaternion separately. For datasets with sparse annotated ground truth, we apply the losses on the outputs and pixels with valid ground truth.

We also include two self-supervised losses on rasterized images, rasterized point and motion:

$$L_{\text{self}} = L_{\text{photo}} + w_{\text{smooth}} L_{\text{smooth}}, \tag{7a}$$

$$L_{\text{photo}} = \sum_{u \in \{t, t+1\}} \left( \text{mse}(\hat{\mathbf{I}}_u, \mathbf{I}_u) + w_{\text{lpips}} \text{lpips}(\hat{\mathbf{I}}_u, \mathbf{I}_u) \right), \tag{7b}$$

$$L_{\text{smooth}} = \sum_{u \in \{t, t+1\}} \sum_{\mathbf{d} \in \{\mathbf{X}, \mathbf{V}\}} \left( |\partial_x \mathbf{d}_u| e^{-|\partial_x \mathbf{I}_u|} + |\partial_y \mathbf{d}_u| e^{-|\partial_y \mathbf{I}_u|} \right), \tag{7c}$$

where photometric loss applies mean square error (MSE) and LPIPS (Zhang et al., 2018) loss between input images ($\mathbf{I}_t$ and $\mathbf{I}_{t+1}$) and respective rasterized images ($\hat{\mathbf{I}}_t$ and $\hat{\mathbf{I}}_{t+1}$). The smoothness loss that encourages spatial smoothness on predicted outputs is applied at rasterized motion, $\mathbf{V}_t$ and $\mathbf{V}_{t+1}$, and rasterized point maps, $\mathbf{X}_t$ and $\mathbf{X}_{t+1}$, based on an edge-aware smoothness loss (Godard et al., 2017; 2019). We find that the smoothness loss effectively clean up floaters in rasterized point and motion.

**Downstream 2D/3D tasks.** By default, our method estimates outputs for 3D perception tasks, *i.e.*, point, 3D scene flow, and camera poses. From them, 2D perception tasks are naturally solved by projecting them into 2D space. Depth is the last channel of estimated point, optical flow is obtained by projecting 3D scene flow, and moving objects are segmented by thresholding the scene flow.

**4D interpolation.** The dynamic nature of our representation enables full 4D interpolation. Given our estimated set of dynamic 3D Gaussians, our method can interpolate image, point, and motion between two time steps at any camera view point inbetween. As described in Eq. (2), interpolated 3D scene can be represented as $\mathcal{G}(t')$ by translating each 3D Gaussians by its motion. We can then simply rasterize image, point, and motion at any view based on the rasterization Eqs. (3) and (4b).

## 4 EXPERIMENTS

**Implementation details.** For training, we use a mixture of Stereo4D (Jin et al., 2025), PointOdyssey (Zheng et al., 2023), and Virtual KITTI 2 (Gaidon et al., 2016) by sampling each

Table 1: **Geometry estimation**: We report end-point error (EPE) for pointmap accuracy and absolute relative error (Abs. Rel.) and $\delta<1.25$ for depth accuracy. Lower is better for EPE and Abs. Rel., and higher is better for $\delta<1.25$.

| Method | Stereo4D | | | Bonn | | | KITTI | | | Sintel final | | |
|---|---|---|---|---|---|---|---|---|---|---|---|---|
| | EPE | Abs. Rel. | $\delta<1.25$ | EPE | Abs. Rel. | $\delta<1.25$ | EPE | Abs. Rel. | $\delta<1.25$ | EPE | Abs. Rel. | $\delta<1.25$ |
| MASt3R (Leroy et al., 2024) | 1.887 | 0.314 | 66.50 | 0.487 | 0.174 | 76.49 | **2.304** | **0.087** | 92.04 | 3.997 | 0.366 | 61.03 |
| MonST3R (Zhang et al., 2025a) | 1.504 | 0.191 | 74.75 | 0.188 | **0.062** | **96.22** | 2.797 | 0.107 | 87.61 | **3.607** | 0.341 | 57.61 |
| DynaDUSt3R (Jin et al., 2025) | 0.811 | 0.112 | 86.07 | 0.424 | 0.084 | 92.75 | 4.889 | 0.116 | 83.39 | 4.362 | **0.301** | 61.04 |
| ZeroMSF (Liang et al., 2025b) | 1.662 | 0.220 | 76.18 | 0.208 | 0.079 | 91.58 | 3.392 | 0.107 | **92.28** | 3.637 | 0.333 | **63.86** |
| St4RTrack (Feng et al., 2025) | 1.467 | 0.209 | 71.26 | 0.181 | 0.067 | 95.17 | 4.731 | 0.125 | 82.86 | 4.145 | 0.357 | 57.14 |
| **UFO-4D (Ours)** | **0.659** | **0.106** | **88.38** | **0.162** | 0.064 | 96.04 | 2.568 | 0.090 | 88.92 | 3.866 | 0.319 | 61.94 |

Table 2: **Motion estimation.** We evaluate the scene flow accuracy on the Stereo4D test split (Jin et al., 2025) and KITTI Scene Flow 2015 Training (Menze et al., 2018). Our approach substantially outperforms others, achieving the best numbers on all metrics.

| Method | Stereo4D | | | | KITTI | |
|---|---|---|---|---|---|---|
| | forward | | backward | | forward | |
| | $EPE_{3D} \downarrow$ | $\delta_{3D}^{0.05} \uparrow$ | $EPE_{3D} \downarrow$ | $\delta_{3D}^{0.05} \uparrow$ | $EPE_{3D} \downarrow$ | $\delta_{3D}^{0.05} \uparrow$ |
| DynaDUSt3R (Jin et al., 2025) | 0.175 | 51.46 | 0.161 | 52.16 | 0.463 | 1.09 |
| ZeroMSF (Liang et al., 2025b) | 0.164 | 45.63 | – | – | 0.442 | 16.20 |
| St4RTrack (Feng et al., 2025) | 0.373 | 26.62 | – | – | 0.751 | 4.01 |
| **UFO-4D (Ours)** | **0.049** | **83.06** | **0.050** | **82.67** | **0.137** | **47.80** |

dataset in a minibatch with probability of 60%, 20%, and 20% respectively. All datasets include (sparse) annotations of 3D points, 3D sceneflow, and camera pose. For efficient storage, we use 10% of the full Stereo4D dataset by subsampling frames of the dataset. We initialize our network (Fig. 2) using pretrained weights from NoPoSplat (Ye et al., 2025) (Gaussian head) and MASt3R (Leroy et al., 2024) (rest), except for the camera head, which is trained from scratch. We train the model with varying aspect ratios with image width to be 512. We apply photometric augmentations and geometric augmentations such as random scale crop, horizontal flip, and temporal flip. We train the model for 120k iteration steps with a global minibatch size of 16. The training takes around three days on four NVIDIA A100 40GB GPUs.

## 4.1 EVALUATION ON 2D AND 3D TASKS

We evaluate UFO-4D on various 2D and 3D geometry and motion tasks, such as depth, point, and scene flow, and compare it with recent methods that are specifically designed for joint geometry and motion estimation given a pair of input frames: DynaDUSt3R (Jin et al., 2025), ZeroMSF (Liang et al., 2025b), and the concurrent work ST4RTrack (Feng et al., 2025).[1]

All methods are evaluated under the same protocol, following DynaDUSt3R and ZeroMSF that use per-image median scale alignment for both point and scene flow. DynaDUSt3R uses the full Stereo4D dataset for training while ZeroMSF uses a mixture of synthetic datasets, including SHIFT, Dynamic Replica, Virtual KITTI 2, MOVi-F, Spring, and PointOdyssey. Please see details on the inference setup of each method in Section A.2.

**Geometry estimation.** Table 1 compares the accuracy of pointmaps and depth for geometry estimation on multiple benchmarks including Stereo4D test split (Jin et al., 2025), Bonn (Palazzolo et al., 2019), and KITTI Scene Flow 2015 Training (Menze et al., 2018). For pointmap, we report averaged end-point error (EPE) between predicted pointmap and ground truth pointmap defined at the canonical camera coordinate. For depth, we report standard depth metrics, absolute relative error (Abs. Rel.) and percentage of inlier points ($\delta<1.25$).

Our method substantially outperforms all direct competitors on Stereo4D and remains very competitive on the rest datasets. The trade-off across datasets of each method can stem from the mixture of training datasets that the method uses (*cf*. Table C). For example, methods (*e. g*. MonST3R and

---

[1]St4RTrack is primarily optimized for long-term tracking, unlike the short-term dynamics evaluated here.

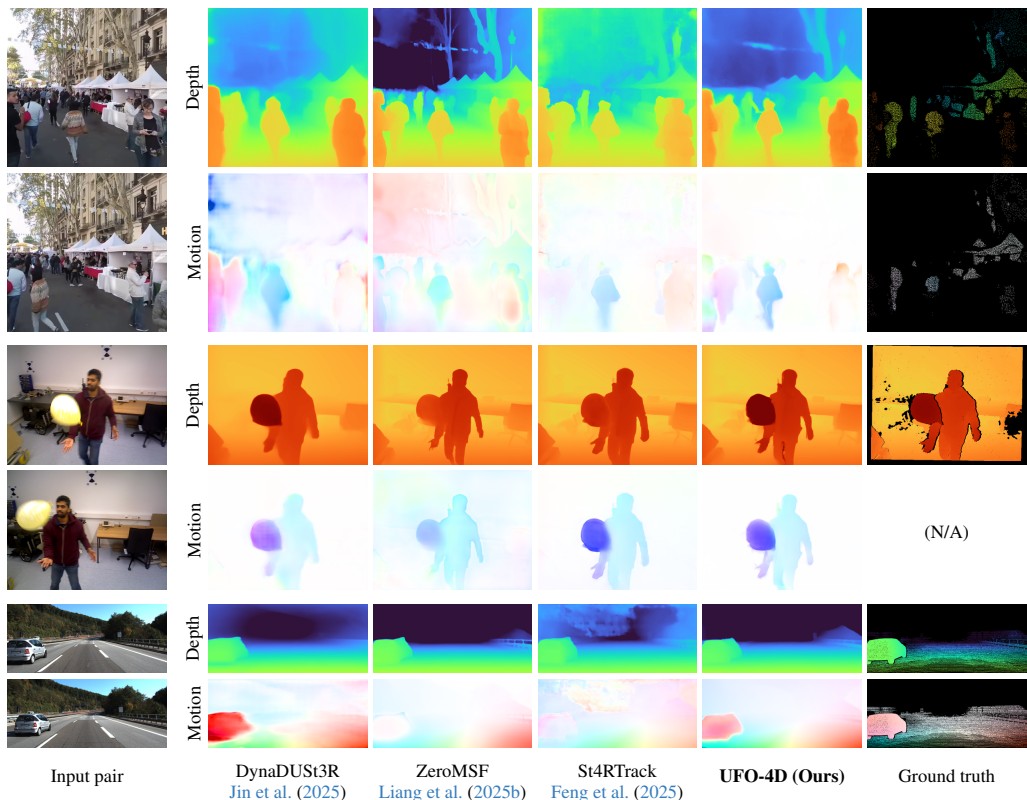

| Input pair | DynaDUSt3R Jin et al. (2025) | ZeroMSF Liang et al. (2025b) | St4RTrack Feng et al. (2025) | **UFO-4D (Ours)** | Ground truth |

Figure 4: **Qualitative comparison** of depth and projected 2D optical flow on Stereo4D, Bonn, and KITTI. For motion on KITTI, it visualizes motion relative to the camera, as GT is defined. Unlike DynaDUSt3R, ZeroMSF and St4RTrack, which suffers from residual motions in static region and inaccurate motion on object boundaries, UFO-4D exhibits clear motion boundaries and separation between moving objects and background. More qualitative results are in Section A.1.

Table 3: **Pose estimation.** We report standard metrics, ATE and RPE translation ($RPE_{trans}$) and rotation ($RPE_{rot}$) for pose estimation on Stereo4D test, Bonn, and Sintel final, Our feed-forward approach outperforms methods relying on iterative solvers (*e. g.* MonST3R, St4RTrack) on all metrics.

| Method | Stereo4D (Jin et al., 2025) | | | Bonn (Palazzolo et al., 2019) | | | Sintel (Butler et al., 2012) | | |
|---|---|---|---|---|---|---|---|---|---|
| | ATE | $RPE_{trans}$ | $RPE_{rot}$ | ATE | $RPE_{trans}$ | $RPE_{rot}$ | ATE | $RPE_{trans}$ | $RPE_{rot}$ |
| MonST3R (Zhang et al., 2025a) | 0.0458 | 0.0647 | 0.805 | 0.0031 | 0.0043 | 0.544 | 0.0290 | 0.0410 | 1.080 |
| St4RTrack (Feng et al., 2025) | 0.0602 | 0.0851 | 0.729 | 0.0024 | 0.0034 | 0.400 | 0.0231 | 0.0326 | 0.884 |
| **UFO-4D (Ours)** | **0.0101** | **0.0142** | **0.179** | **0.0020** | **0.0028** | **0.271** | **0.0122** | **0.0172** | **0.298** |

ZeroMSF) that use the Spring dataset for training usually show good performance on Sintel due to their narrow domain gap. Fig. 4 visualizes qualitative results of all methods. Our method estimates good depth boundaries especially where supervision signal is missing.

**Scene flow estimation.** Table 2 compares with methods that estimate scene flow between two monocular images. The metrics includes 3D end-point-error (EPE) between ground truth and predicted estimates and the fraction of 3D points that have motion error within 5 cm ($\delta_{3D}^{0.05}$) compared to ground truth. Our method achieves the best accuracy on all metrics, especially with more than three times lower EPE on Stereo4D and KITTI than the best method. Fig. 4 visualizes the source of the gain of our method. Unlike other methods, our method shows clear motion boundaries and disentanglement of motion between background and moving objects.

**Pose estimation.** Table 3 evaluates pose accuracy on Stereo4D test split, Bonn, and Sintel Final. Given two-frame inputs, we estimate relative pose and report accuracy using standard metrics, in-

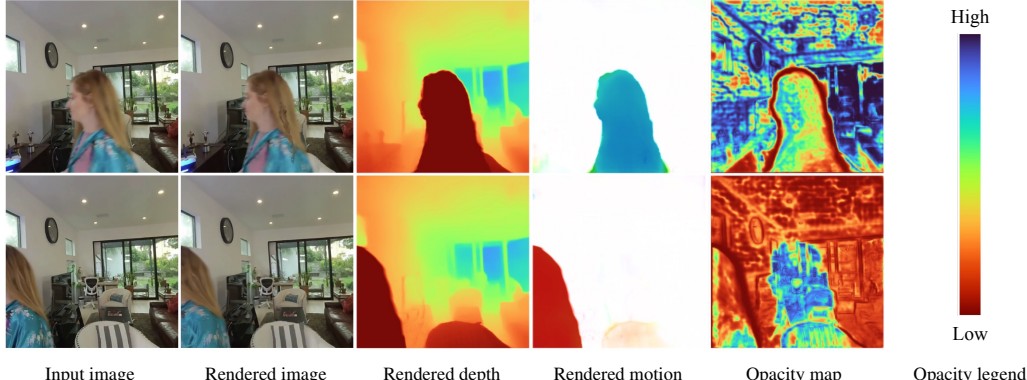

| Input image | Rendered image | Rendered depth | Rendered motion | Opacity map | Opacity legend |

Figure 5: **Opacity as learnable confidence**: Opacity maps show the model's behavior in a (dis)occlusion scenario. Our model learns to assign high confidence (opacity) to disoccluded regions, and for mutually-visible regions, it selects only one corresponding Gaussian from the two views, enabling an efficient and compact 4D representation.

cluding ATE and RPE (translation and rotation). We provide more details on evaluation protocol in Appendix. Our method achieves the best performance across all datasets. Unlike MonST3R (Zhang et al., 2025a) and St4RTrack Feng et al. (2025), which obtain pose via a PnP solver (Lepetit et al., 2009) with RANSAC (Fischler & Bolles, 1981), our approach is fully feed-forward. We demonstrate that direct estimation eliminates the need for additional post-processing while simultaneously achieving higher accuracy. Further analysis in Section A.10 using PnP+RANSAC on our method confirms that our gain originates from both more accurate geometry and the robustness of our feed-forward estimator.

## 4.2 ANALYSES

**Opacity as learnable confidence.** One notable emergent property of our model is that opacity serves as a learnable confidence metric. Through the defined loss in Eq. (5), our feedforward model learns to predict an opacity value for each Gaussian that properly weight its contribution to the rendered image, points, and motion. Figure 5 visualizes one example where heavy occlusion and disocclusion occurs. The last column visualizes an opacity map for each image, where blue means high opacity and therefore large contributions to rendered outputs. We see that not all Gaussians have high opacity: For regions that are visible in both views (*e. g.* background and the person), the model chooses one of the corresponding Gaussians from the two images to compactly represent the scene. Regions that are occluded in one view and become visible in the other are also highlighted as high opacity (*e. g.* background being occluded by the person in the second view). This emergent property allows the model to autonomously attenuate the influence of less relevant Gaussians for efficient 4D scene representation.

**Loss ablation.** In Table 4, we analyze how using 3D Gaussians as a representational bottleneck encourages synergy between tasks during joint training. We train the models on the Stereo4D dataset at a $256 \times 256$ resolution for 40k iterations with a batch size of 12. We report image reconstruction (PSNR) and the accuracy of both Gaussian attributes (*i. e.*, center and velocity) and rasterized outputs for points and motion (EPE).

When gradients from the photometric loss to the Gaussian center and velocity are removed ((a) → (b)), point and motion accuracy drops significantly, highlighting the importance of our photometric loss (Eq. (7c)). Also, removing the supervision on rasterized points and motion ((a) → (c)), results in a substantial accuracy degradation across all tasks. Qualitative examples in Fig. 6 visualize these findings. The rendering losses for points and motion are crucial for sharp motion boundaries, which tells its significant more than numbers in the table.

A model without any rendering losses (d), *i. e.*, only estimating 3D point and motion of each pixel, performs behind our full model. This indicates that even though our network shares its capacity to

Table 4: **Loss ablation.** We report image reconstruction in PSNR and accuracy of Gaussian center (Point), rasterized point (Point rast.), Gaussian velocity (Motion), and rasterized motion (Motion rast.) in end-point error (EPE). Integrating the photometric loss gradient (Eq. (7c)) with all heads boosts overall performance. Losses on rendered motion (Eq. (6b)) and point map (Eq. (6c)) are crucial for achieving high accuracy.

| Alias | Configuration | Image (PSNR ↑) | Point (EPE ↓) | Point rast. (EPE ↓) | Motion (EPE ↓) | Motion rast. (EPE ↓) |
|---|---|---|---|---|---|---|
| (a) | Full model | 23.929 | **0.827** | **0.841** | **0.069** | **0.064** |
| (b) | W/o image gradient. | **24.268** | 0.903 | 0.899 | 0.072 | 0.070 |
| (c) | W/o motion and point rendering | 20.675 | 0.911 | 0.943 | 0.071 | 0.069 |
| (d) | W/o motion, point, and image rendering | – | 0.886 | – | 0.071 | – |

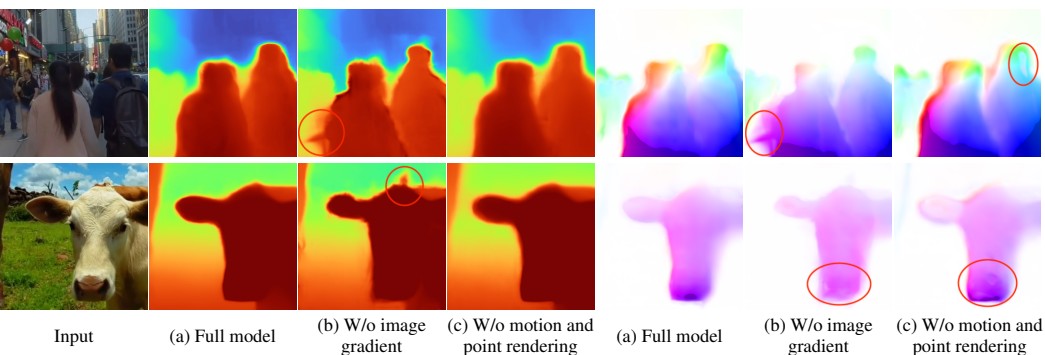

| Input | (a) Full model | (b) W/o image gradient | (c) W/o motion and point rendering | (a) Full model | (b) W/o image gradient | (c) W/o motion and point rendering |

Figure 6: **Qualitative comparison on loss ablation.** Gradient backpropagation from the image synthesis loss and rendering losses on point and motion helps UFO-4D improve motion and point estimates, especially on object and motion boundaries.

Table 5: **Architecture comparison.** For apples-to-apples comparison, we train other methods with different output representation on our training protocol. Our method achieves the best accuracy except for Bonn on both point and motion end-point errors (lower the better).

| Output representation | Equivalent or closest models | Stereo4D | | KITTI | | Bonn |
|---|---|---|---|---|---|---|
| | | Motion EPE | Point EPE | Motion EPE | Point EPE | Point EPE |
| Dynamic 3DGS | **UFO-4D (Ours)** | 0.058 | **0.781** | **0.244** | **3.553** | 0.211 |
| Per-pixel point and motion | DynaDUSt3R, ZeroMSF | **0.057** | 0.809 | 0.283 | 4.567 | **0.196** |
| Per-pixel point | MonST3R | – | 0.804 | – | 4.562 | 0.201 |

estimating Gaussian attributes, the supervision enabled by our differentiable 4D rasterizer leads to better point and motion accuracy. Further, this approach unlocks novel capabilities, 4D interpolation.

**Architecture comparison.** As methods often rely on distinct training recipes, it is crucial to decouple architectural benefits from implementation details. To achieve this, we compare our method with other baseline methods trained under our training protocol for apples-to-apples comparison. We change output representation to *(i)* per-pixel point and motion estimation (equivalent to DynaDUSt3R (Jin et al., 2025) and ZeroMSF (Liang et al., 2025b)) and *(ii)* per-pixel point only (equivalent to MonST3R). We follow the original training setup except that we use $256 \times 256$ training resolution and train model for 60k iteration steps.

Table 5 shows that our method outperforms per-pixel representations on Stereo4D and KITTI, with substantial gains on KITTI. However, our performance on Bonn is slightly behind. We attribute this to the prevalence of textureless regions in Bonn where our model predicts large-scale, overlapping Gaussians (Fig. E). While this ensures spatial continuity, it introduces error-prone spatial dependencies among neighboring pixels unlike per-pixel point representation. More discussion is followed in Section A.7.

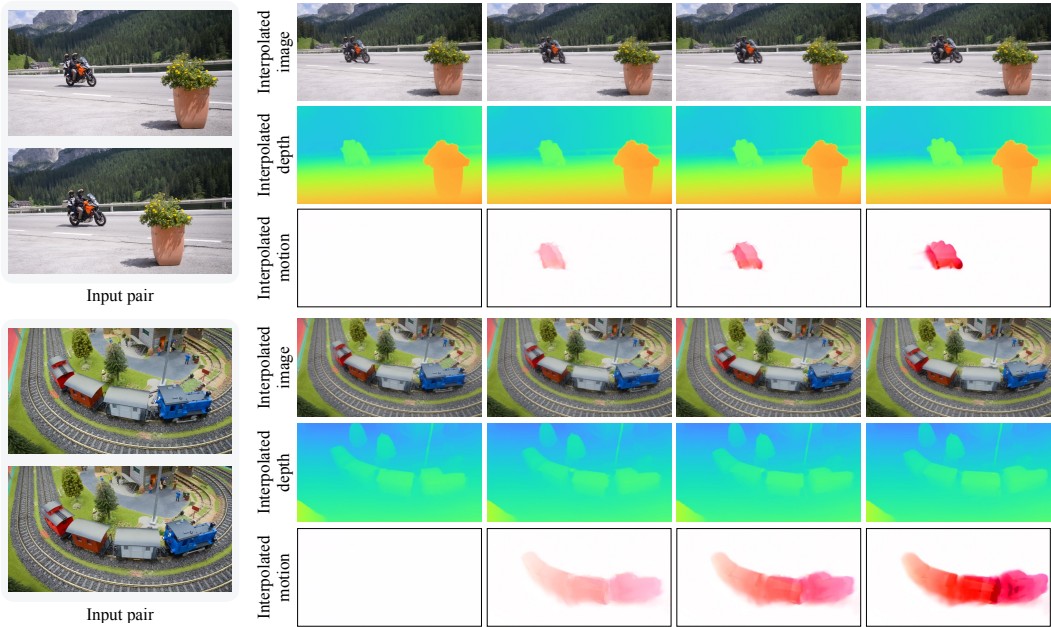

Figure 7: **4D interpolation** on DAVIS. Given an input pair on the left, the righthand side shows rasterized images, depth, and motion maps at the canonical camera coordinate at interpolated time between the two input frames. Our method also clearly segments out moving objects.

**4D interpolation.** We demonstrates the new capability of our method: 4D interpolation. Given an input pair, Fig. 7 shows rendered image, depth, and motion maps at the canonical camera coordinate at interpolated time between the two input frames, via 4D differentiable rasterizer. Our method can render those at any time at any camera views.

## 5 CONCLUSION

We have introduced UFO-4D, a unified feedforward model for 4D reconstruction from a pair of unposed images. By predicting a holistic dynamic 3D Gaussian Splatting representation, UFO-4D effectively leverages self-supervision to overcome challenges like data scarcity and occlusions. UFO-4D significantly outperforms prior work in estimating 3D geometry and motion, while uniquely enabling high-fidelity 4D spatio-temporal interpolation. Our work demonstrates the power of unified, explicit representations for dynamic scene understanding, opening promising avenues for extending this framework to long-sequence videos and more complex motion models.

**Future work.** We identify several directions for future explorations. A naive extension to long sequences would be memory-intensive due to the linear growth of Gaussian Splats; future work could explore a compact scene representation (An et al., 2026; Xu et al., 2025a). Also, our model's assumption of linear motion and constant brightness is best suited for short time intervals. This could be addressed by incorporating learnable, non-linear motion models and time-varying Gaussian attributes to handle more complex dynamics and photometric changes. Moreover, rendered images, motion, and geometry at novel view and time can be included as additional supervision objectives to enhance spatio-temporal consistency, when their annotations (*e. g.* from multi-view video datasets) are available.

### ACKNOWLEDGMENTS

We sincerely thank Richard Szeliski, Junyi Zhang, Michael Rubinstein, Richard Tucker, and Linyi Jin for their helpful discussions and technical assistance throughout the project. We also appreciate the anonymous reviewers for their constructive feedback during the review process.

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

## A  APPENDIX

### A.1  ADDITIONAL QUALITATIVE COMPARISON

We present additional qualitative results on Stereo4D (Fig. A), KITTI (Fig. B), and Bonn (Fig. C). Our method consistently shows more accurate depth and motion estimates than competing approaches. A notable advantage is our robustness to significant camera rotation and object motion (*e. g.* the first two rows in Fig. A), where other methods often produce erroneous estimates. Furthermore, our method effectively disentangles dynamic object motion from camera ego-motion, successfully isolating moving objects where other methods frequently assign non-zero motion to static backgrounds. Unlike the blurred predictions seen in DynaDUSt3R, our method preserves sharp motion boundaries and geometric consistency across all datasets.

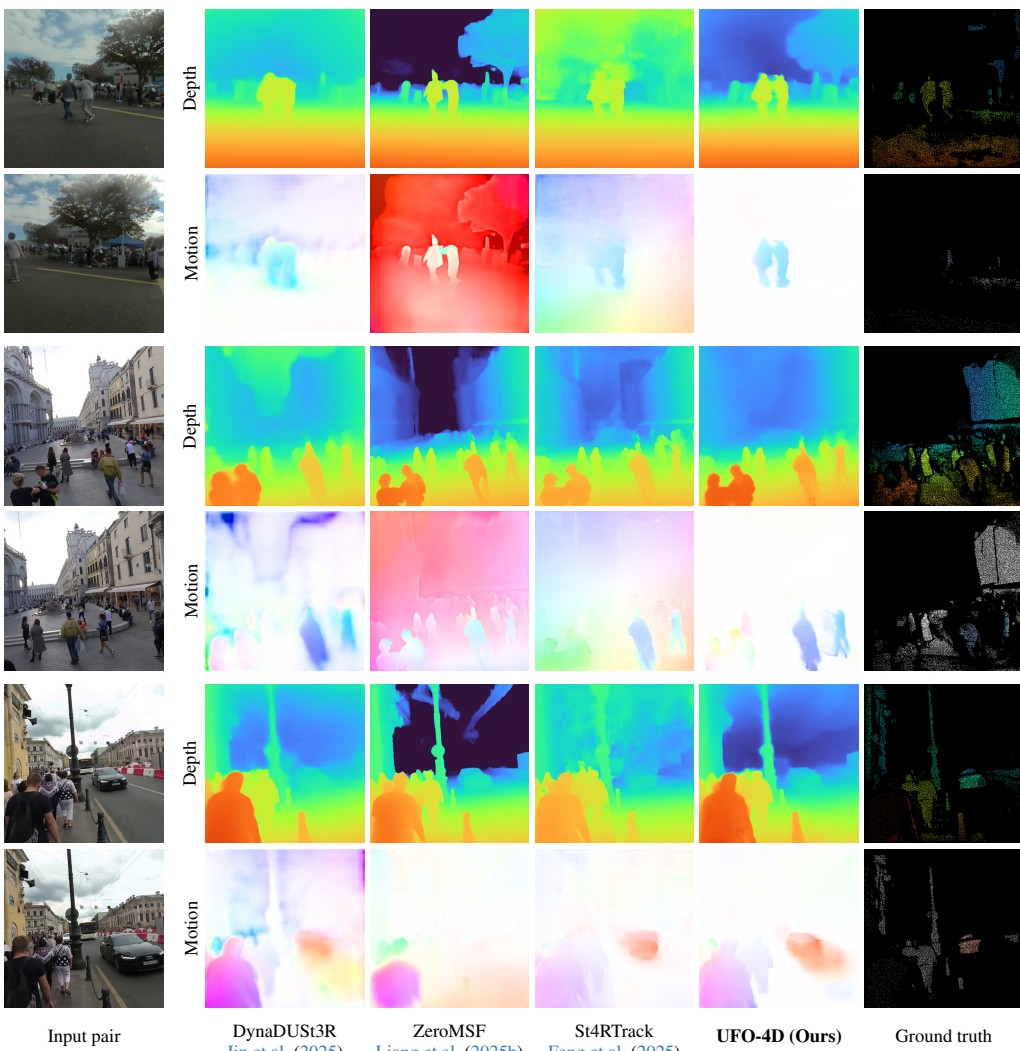

Figure A: **Qualitative comparison on Stereo4D.** We visualize depth and projected 2D optical flow. UFO-4D consistently outperforms other methods in depth and motion accuracy. Notably, in scenes with significant camera rotation (top two rows), our method robustly estimates 3D geometry and isolates 3D motion of dynamic objects in the canonical space. In contrast, other methods struggle to disentangle camera ego-motion, resulting in residual motion artifacts on static background.

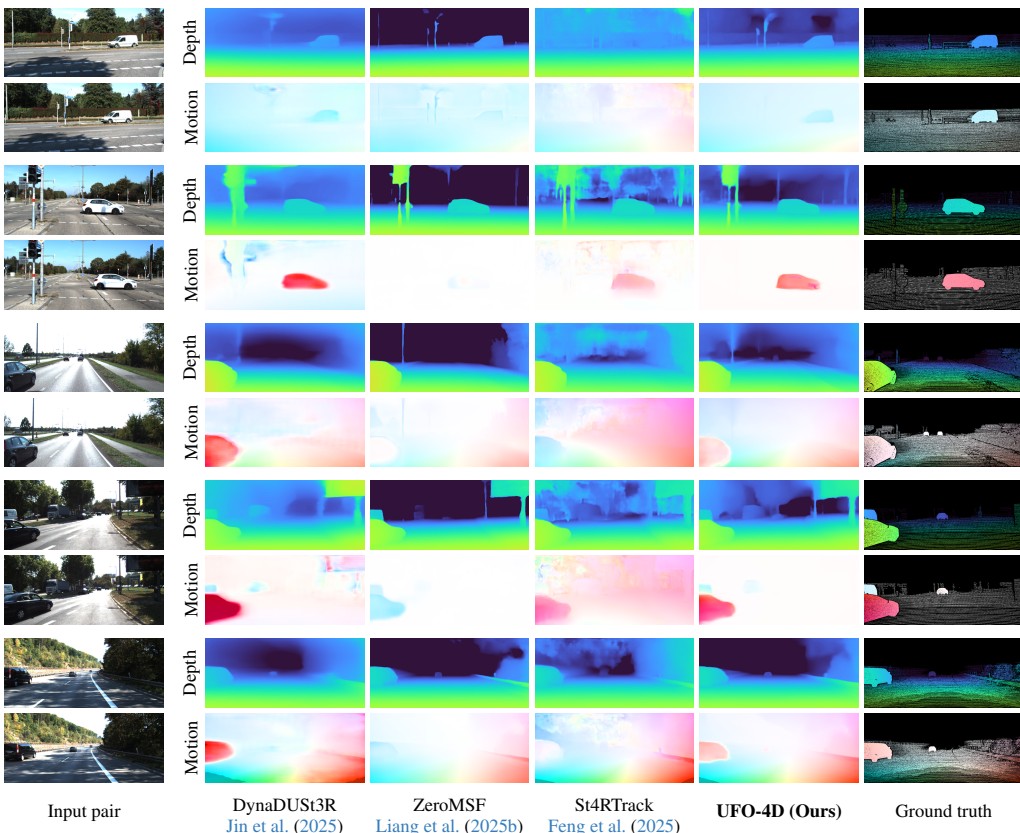

Figure B: **Qualitative comparison on KITTI.** We visualize motion in camera coordinates, consistent with the ground truth of KITTI. Our method outperforms competitors with notably clearer motion boundaries, whereas ZeroMSF and St4RTrack frequently show erroneous motion estimation.

## A.2 EVALUATION PROTOCOL

In Section 4.1, we compare our method with direct competitors, *e.g.*, DynaDUSt3R (Jin et al., 2025), ZeroMSF (Liang et al., 2025b), and St4RTrack (Feng et al., 2025), that output point and 3D scene flow. Due to that each method has their own inference setups, we include details here on their evaluation setups used in our comparison study.

**DynaDUSt3R.** Similar to our method, DynaDUSt3R (Jin et al., 2025) outputs pointmaps and scene flow in the first frame's coordinate system, allowing us to use its predictions directly. For inference, we resize inputs to $512 \times 512$ for Stereo4D and Bonn. For KITTI and Sintel, we use resize inputs to $382 \times 512$ following their protocol and $208 \times 512$ preserving aspect ratio, respectively. All outputs are resized back to the original resolution for evaluation.

**ZeroMSF.** We use the same inference resolution as in DynaDUSt3R (Jin et al., 2025). While ZeroMSF (Liang et al., 2025b) defines pointmaps in the first camera coordinate, it parameterizes scene flow as Camera-space 3D offsets (CSO)—estimating motion relative to a fixed camera. To ensure fair comparison on datasets where their motion is defined at the canonical camera coordinate (Stereo4D), we convert this flow to world coordinates using ground truth extrinsics. Although using ground truth poses grants ZeroMSF an advantage, it is necessary to minimize errors arising from coordinate system mismatch.

**MASt3R, MonST3R, and St4RTrack.** For MASt3R, MonST3R, and St4RTrack, we follow the original evaluation protocol in their implementation. Unlike our method, MonST3R and St4RTrack

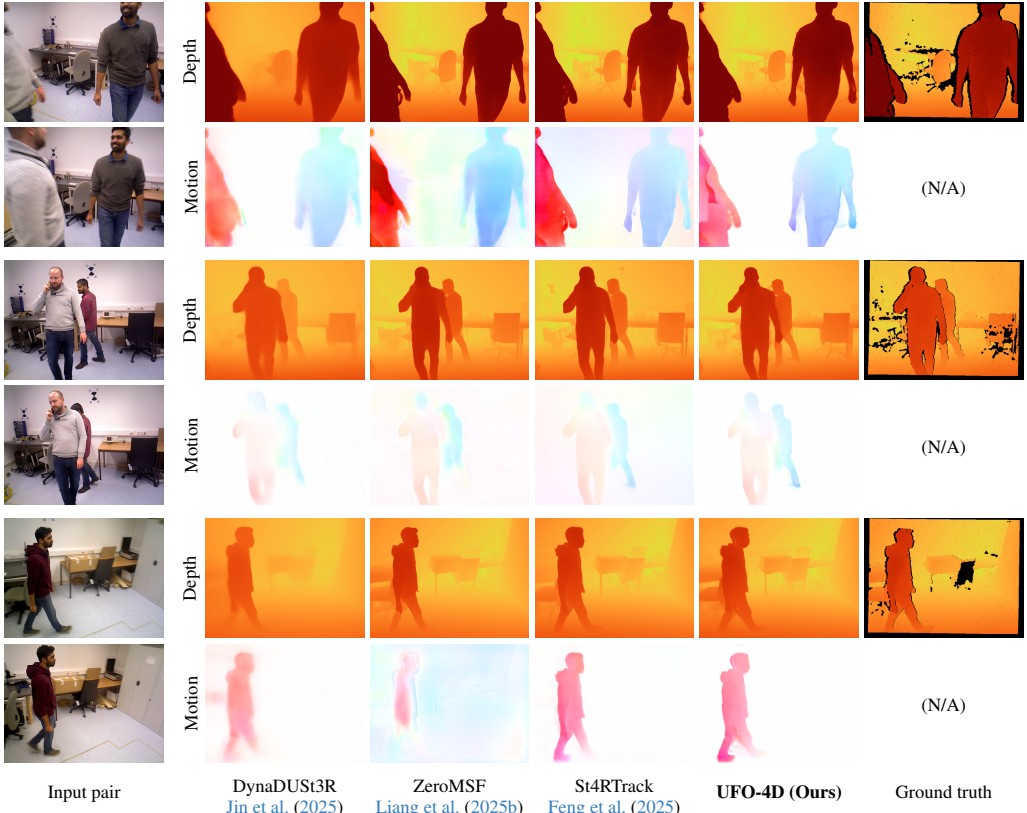

Figure C: **Qualitative comparison on Bonn.** Comparing to other methods, our method outputs clear 3D motion disentangled from camera motion. ZeroMSF and St4RTrack suffer from non-zero residual motion in background. DynaDUSt3R tends to show unclear motion boundaries.

do not directly output relative camera pose in a feedforward manner. For the pose estimation, we use a PnP solver with RANSAC with their default hyperparameters.

### A.3 SOME OBSERVATIONS ABOUT THE STEREO4D DATASET

Stereo4D (Jin et al., 2025) provides a large-scale annotated data for 4D understanding, specifically including camera intrinsics, extrinsics, depth, 3D point trajectory. While it is the biggest real-world based annotated dataset, we found that it includes noisy annotations that residuals from its optimization process given depth and motion cues processed from off-the-shelf models. Fig. D visualizes some samples from Stereo4D. The last column visualizes motion map in the standard optical flow color coding. Close examination shows that some static region (*e. g.*, ground and building wall) include non-zero motion, which is typically under centimeter level.

This noisy annotation may result in non-zero motion estimates for static parts, as visualized in Fig. 4 for DynaDust3R. Our dense, self-supervised photometric loss on rendered images complements the imperfect ground truth and lead to better motion accuracy than other methods.

### A.4 EVALUATION WITH PER-VALID-PIXEL AVERAGING

While the main paper reports metrics using per-frame averaging (where each frame is weighted equally), an alternative protocol used by DynaDUSt3R (Jin et al., 2025) is per-valid-pixel averaging. This approach weights each frame's contribution by its number of valid pixels. For a comprehensive comparison, we report results using this metric as well. Under this per-valid-pixel evaluation, UFO-4D also consistently outperforms other baselines on Stereo4D and KITTI and perform very competitive on Bonn and Sintel final, as shown in Tables A and B.

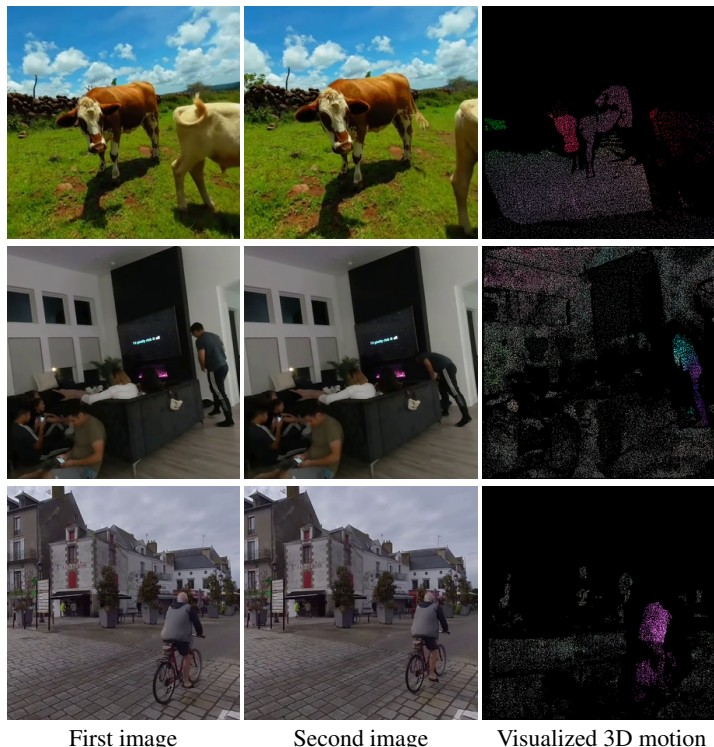

| First image | Second image | Visualized 3D motion |

Figure D: **Visualization of annotation of Stereo4D.** The first two columns visualize two input frames, and the last columns visualizes 3D motion annotation defined at the canonical camera co-ordinate. While Stereo4D provides large-scale real-world annotated data, the label includes noisy annotation, *e. g.*, non-zero 3D motion for static region.

Table A: **Geometry estimation (under per-valid-pixel averaging protocol)**: Using the same evaluation metric in DynaDUSt3R, our method consistently outperforms other methods.

| Method | Stereo4D | | | Bonn | | | KITTI | | | Sintel final | | |
|---|---|---|---|---|---|---|---|---|---|---|---|---|
| | EPE | Abs. Rel. | $\delta<1.25$ | EPE | Abs. Rel. | $\delta<1.25$ | EPE | Abs. Rel. | $\delta<1.25$ | EPE | Abs. Rel. | $\delta<1.25$ |
| MASt3R (Leroy et al., 2024) | 1.658 | 0.290 | 65.60 | 0.484 | 0.173 | 76.68 | **2.247** | **0.086** | 92.15 | 3.907 | 0.360 | 60.68 |
| MonST3R (Zhang et al., 2025a) | 1.380 | 0.194 | 72.57 | 0.187 | **0.062** | **96.26** | 2.741 | 0.106 | 87.96 | **3.514** | 0.338 | 57.23 |
| DynaDUSt3R (Jin et al., 2025) | 0.683 | **0.103** | 87.94 | 0.423 | 0.084 | 92.83 | 4.805 | 0.113 | 84.10 | 4.296 | **0.295** | 60.88 |
| ZeroMSF (Liang et al., 2025b) | 1.674 | 0.241 | 72.77 | 0.207 | 0.078 | 91.68 | 3.265 | 0.103 | **92.66** | 3.560 | 0.327 | **63.80** |
| St4RTrack (Feng et al., 2025) | 1.269 | 0.198 | 71.37 | 0.180 | 0.067 | 95.24 | 4.661 | 0.124 | 83.13 | 4.097 | 0.349 | 56.87 |
| **UFO-4D (Ours)** | **0.594** | 0.103 | **88.20** | **0.162** | 0.064 | 96.08 | 2.485 | 0.088 | 89.41 | 3.828 | 0.317 | 61.45 |

Table B: **Motion estimation (under per-valid-pixel averaging protocol)** We evaluate the scene flow accuracy on the Stereo4D test split (Jin et al., 2025) and KITTI Scene Flow 2015 Training (Menze et al., 2018). Our approach consistently outperforms others on all metrics.

| Method | Stereo4D | | | | KITTI | |
|---|---|---|---|---|---|---|
| | forward | | backward | | forward | |
| | $EPE_{3D}$ | $\delta_{3D}^{0.05}$ | $EPE_{3D}$ | $\delta_{3D}^{0.05}$ | $EPE_{3D}$ | $\delta_{3D}^{0.05}$ |
| DynaDUSt3R (Jin et al., 2025) | 0.118 | 65.19 | 0.106 | 64.47 | 0.467 | 1.09 |
| ZeroMSF (Liang et al., 2025b) | 0.067 | 67.38 | – | – | 0.452 | 13.84 |
| St4RTrack (Feng et al., 2025) | 0.233 | 42.56 | – | – | 0.768 | 3.36 |
| **UFO-4D (Ours)** | **0.023** | **91.78** | **0.025** | **91.28** | **0.139** | **46.71** |

Table C: **Dataset ablation.** Given three training datasets, Stereo4D (ST), PointOdyssey (PO), and Virtual KITTI 2 (VK), we try a different mixture of each dataset, train our model, and report performance on three evaluation datasets, Stere4D, KITTI, and Bonn, reporting PSNR for image reconstruction and end-point error (EPE) for point and motion. The best accuracy on Stereo4D is achieved by solely training on Stereo4D. Adding Virtual KITTI2 improves especially motion accuracy on KITTI but hurts point accuracy on Bonn. Adding PointOdyssey improves accuracy on Bonn but hurt that on KITTI as trade-off. All experiment is done in a small scale training.

| Dataset proportion | | | Stereo4D | | | KITTI | | | Bonn | |
|---|---|---|---|---|---|---|---|---|---|---|
| ST | PO | VK | Image ($\uparrow$) | Point ($\downarrow$) | Motion ($\downarrow$) | Image ($\uparrow$) | Point ($\downarrow$) | Motion ($\downarrow$) | Image ($\uparrow$) | Point ($\downarrow$) |
| 0.7 | 0.15 | 0.15 | 23.547 | 0.065 | 0.891 | 18.845 | 0.273 | 3.427 | 18.568 | 0.314 |
| 0.85 | – | 0.15 | 23.419 | 0.065 | **0.839** | 18.791 | 0.298 | 4.102 | 18.056 | 0.344 |
| 0.85 | 0.15 | – | 23.587 | **0.064** | 0.874 | 18.037 | 0.355 | 6.214 | 18.610 | 0.291 |
| – | 0.5 | 0.5 | 21.918 | 0.074 | 1.292 | **19.363** | **0.267** | **2.761** | **19.824** | **0.273** |
| 1.0 | – | – | **23.929** | **0.064** | 0.841 | 18.610 | 0.311 | 6.138 | 18.913 | 0.316 |

Table D: **Training with different initialization.** We compare our model trained on different pretrained weights, report image PSNR, 3D point EPE, and 3D motion EPE on Stereo4D (Jin et al., 2025), KITTI (Menze et al., 2018), and Bonn (Palazzolo et al., 2019) datasets. Initialization from MASt3R gives overall better accuracy whereas that from MonST3R improves 3D point accuracy on KITTI and Bonn.

| Method | Stereo4D | | | KITTI | | | Bonn | |
|---|---|---|---|---|---|---|---|---|
| | PSNR$\uparrow$ | Point EPE$\downarrow$ | Motion EPE$\downarrow$ | PSNR$\uparrow$ | Point EPE$\downarrow$ | Motion EPE$\downarrow$ | PSNR$\uparrow$ | Point EPE$\downarrow$ |
| Initialization from MASt3R (**baseline**) | **24.519** | **0.781** | **0.058** | **19.606** | 3.553 | **0.244** | **26.606** | 0.211 |
| Initialization from MonST3R | 24.200 | 0.868 | 0.059 | 19.506 | **3.260** | 0.258 | 26.551 | **0.155** |

## A.5 DATASET ABLATION

To analyze how each dataset combination affects accuracy across differenet evaluation dataset, we conduct an ablation study on different mixture of training datasets. Table C provides the result.

Given three training datasets, Stereo4D (ST), PointOdyssey (PO), and Virtual KITTI2 (VK), we train our full model using the same small-scale training configuration as in Table 4 in the main paper. During training, we sample each dataset given each provided dataset proportion in the table.

Addition of different dataset on top of Stereo4D gives different behaviors on each test dataset. The best accuracy on Stereo4D dataset can be achieved by training solely on the Stereo4D dataset, instead of the mixture of different datasets. The addition of Virtual KITTI2 improves accuracy on KITTI but hurts accuracy on Stereo4D and Bonn as a trade-off. The addition of PointOdyssey does affect accuracy on Stereo4D much, while improving accuracy on Bonn but degrading accuracy on KITTI. This concludes that there is no single best dataset for generalization. A mixture of datasets improves generalization to other datasets in general, but the accuracy obtained is worse than that of the model trained on a dataset specialized to the target domain.

## A.6 MODEL INITIALIZATION

As described in Section 4, we initialize our network using pretrained weights from MASt3R (Leroy et al., 2024), with the Gaussian head initialized from Ye et al. (2025). We also explore initializing from MonST3R (Zhang et al., 2025a) to leverage its capability of dynamic scene reconstruction.

Table D compares these different initializations on the Stereo4D, KITTI, and Bonn datasets using the training setup from the architecture comparison study in Section 4.2 (*i.e.*, $256 \times 256$ training resolution, 60k iteration steps). Interestingly MASt3R initialization yields overall better accuracy. MonST3R provides better 3D point accuracy on Bonn and KITTI. We presume that the performance of the checkpoint from MonST3R is particularly optimized for Bonn, as observed their competitive accuracy on Bonn in Table 1, though our model does not necessarily need this specific prior.

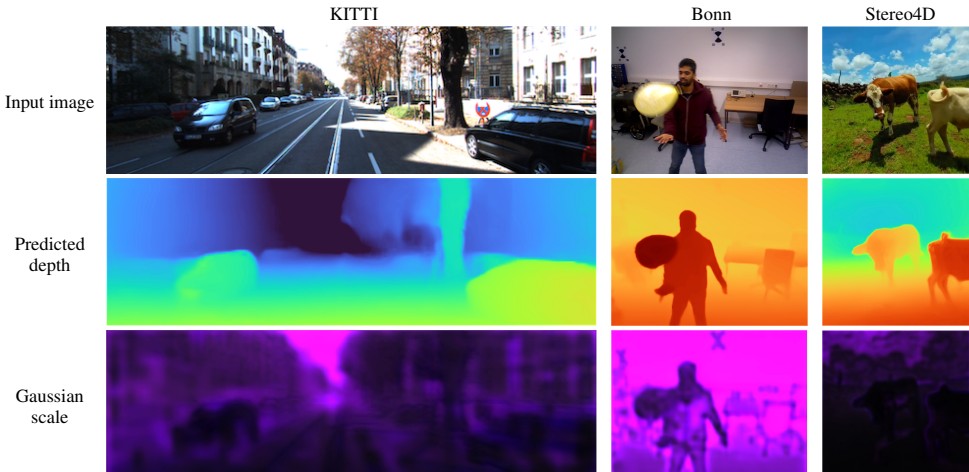

Figure E: **Visualization of the scale of 3D Gaussian.** For each dataset, KITTI, Bonn, and Stereo4D, we visualize Gaussian scale (the bottom row) along with an input image and predicted depth of our method. The Gaussian scale $(x, y, z)$ is visualized in the $(R, G, B)$ color space. For textureless region such as sky or wall, our method predicts larger scale of gaussian whereas small-scale 3D Gaussians are dominant in high-frequency texture region. Bright color means larger Gaussians.

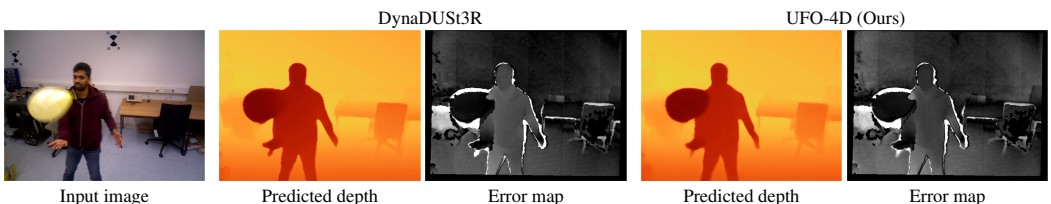

Figure F: **Comparison with DynaDUSt3R on Bonn.** Unlike per-pixel point representation, our representation gives subtle inaccuracy in geometry on textureless region (*e. g.* wall). Brighter means higher errors.

## A.7 DISCUSSION ON BONN

While our method outperforms baselines across most benchmarks, its performance on the Bonn dataset is comparable to prior work (Tables 1 and 5). We attribute this to the prevalence of textureless regions (*e. g.* walls) in Bonn, where our model predicts large, highly overlapping Gaussians (Fig. E). Although this overlap promotes surface continuity, it renders the point estimate dependent on a weighted combination of multiple Gaussians, making it susceptible to accumulated blending errors from nearby Gaussians. In contrast, per-pixel representations (*e. g.* DynaDUSt3R, MonST3R) avoid this spatial dependency by estimating depth in isolation.

Consequently, as visualized in Fig. F, our method exhibits slightly higher errors in these texture-less regions. We quantitatively confirm this hypothesis in Table E, which demonstrates that depth accuracy degrades as the scale of the estimated 3D Gaussians increases.

## A.8 CROSS-ATTENTION MAP VISUALIZATION OF THE CAMERA POSE TOKEN.

Our method is capable of estimating accurate relative camera poses in scenes with dynamic objects. To investigate the underlying mechanism enabling this robustness, we visualize the cross-attention maps between the learnable camera pose token and the image tokens in Fig. G. As a reference, we also visualize a static baseline for each scene by using two identical images as input. We observe that at specific blocks (*e. g.* the 8th, 11th, and 12th blocks), the cross-attention maps exhibit significantly higher attention scores in static regions, effectively suppressing attention on moving objects.

Table E: **Impact of Gaussian scale on geometry accuracy on Bonn.** We report Point EPE and Depth Abs. Rel. across different ranges of estimated Gaussian scales. We observe that depth accuracy decreases along with the increased scale of estimated 3D Gaussians.

| Scale Range | Point (EPE) ↓ | Depth (Abs. Rel.) ↓ |
|---|---|---|
| $0 - 0.5$ | 0.189 | 0.073 |
| $0.5 - 1.0$ | 0.151 | 0.059 |
| $1.0 - 1.5$ | 0.200 | 0.068 |
| $1.5 - 2.0$ | 0.189 | 0.067 |
| $2.0 - 2.5$ | 0.185 | 0.071 |
| $2.5 - 3.0$ | 0.205 | 0.085 |

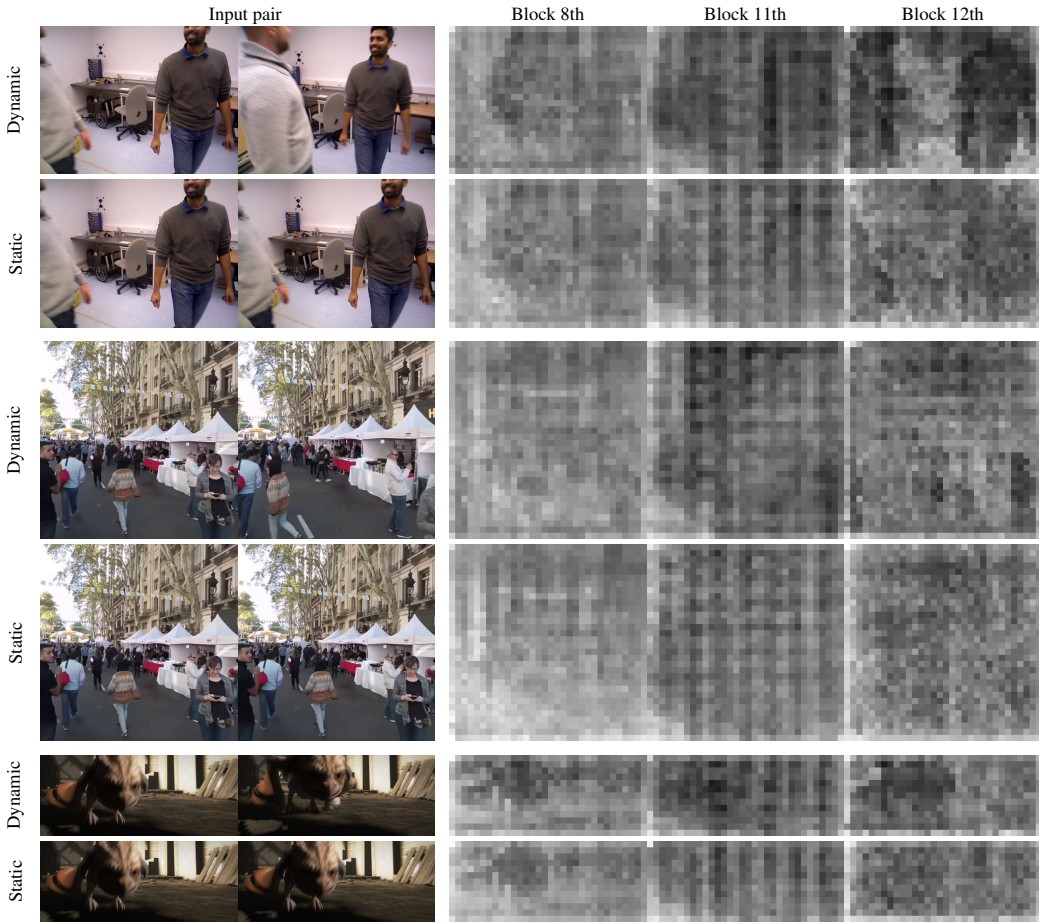

Figure G: **Visualization of cross-attention maps of the camera pose token.** We observe that at specific blocks (Blocks 8th, 11th, and 12th), the camera pose token more attends to image tokens located in static regions (*i. e.*, high attention scores in brighter colors), which provide reliable cues for relative camera pose estimation. Moving objects tend to have darker colors, *i. e.*, lower attention scores. As a reference at each second row, we also visualize the cross-attention maps for a static scene configuration, where identical images are used as input.

## A.9    DISCUSSION ON THE ACCURACY GAIN ON KITTI

In Table 5 in the main paper, our representation shows much better accuracy gain on KITTI, comparing to the model with per-pixel point and motion representation. We find that the huge accuracy gain on KITTI is from both the accurate pose estimation and better supervision with our photometric losses.

Table F: **Ablation on Pose and Representation.** We compare our method with per-pixel baselines using different pose estimators (pose head vs. PnP-RANSAC). Our Dynamic 3DGS representation with the feedforward pose head achieves the best accuracy on Stereo4D and KITTI.

| Output representation | Pose estimator | Stereo4D | | KITTI | | Bonn |
|---|---|---|---|---|---|---|
| | | Motion EPE | Point EPE | Motion EPE | Point EPE | Point EPE |
| Dynamic 3DGS (**Ours**) | Pose head | **0.057** | **0.781** | **0.244** | **3.553** | 0.211 |
| Per-pixel point and motion | Pose head | 0.060 | 0.811 | 0.245 | 4.396 | 0.206 |
| | PnP-RANSAC | 0.058 | 0.809 | 0.283 | 4.567 | **0.196** |

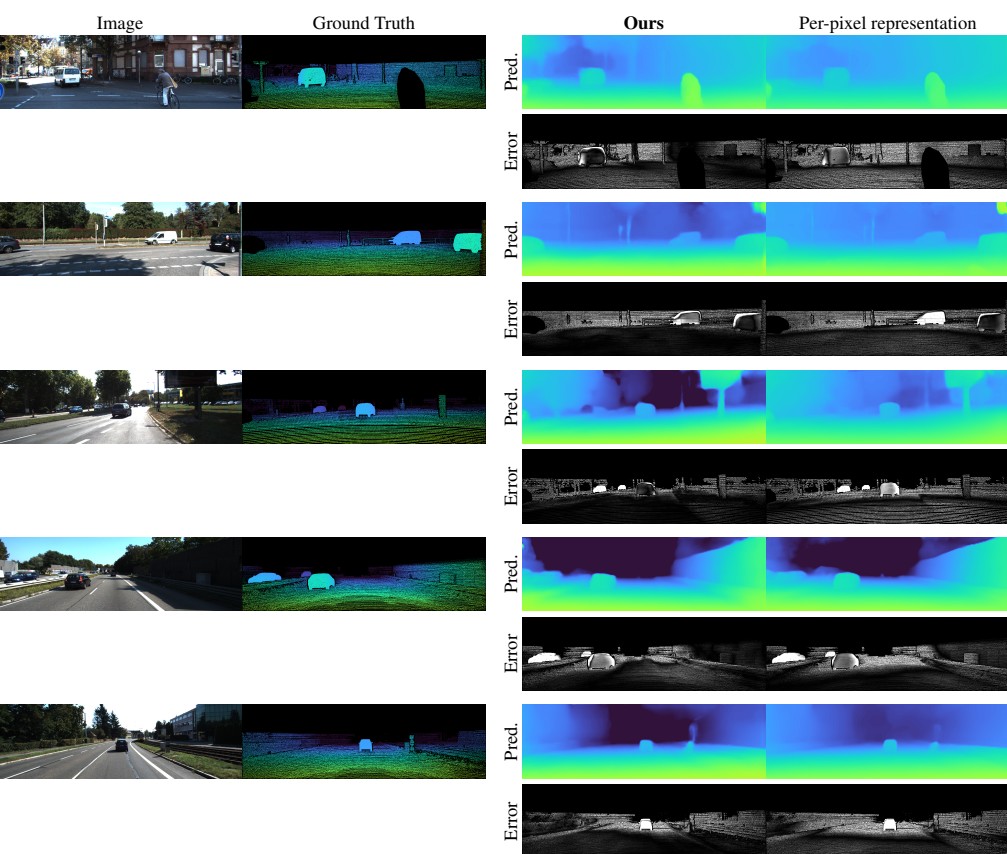

Figure H: **Qualitative comparison of our method and baseline with per-pixel point representation on KITTI.** Unlike per-pixel representation model, our method yields much better accuracy especially on planar road surface, benefiting from our accurate pose estimation and photometric losses on rendered images. In error maps, darker means more accurate.

Our accuracy gain for motion on KITTI primarily originates from our accurate pose estimation. Since KITTI evaluates motion in local camera coordinates, predictions in the canonical camera coordinate need to be transformed using estimated poses. For baselines, we use PnP-RANSAC that follows their formulation. In contrast, Our method utilizes a significantly more accurate, directly estimated feedforward pose, leading better motion accuracy in the end. We validated this source of improvement by training a per-pixel baseline equipped with a pose head (*i.e.* equivalent to DynaDUSt3R with the pose head.). As in Table F, its motion accuracy on KITTI now almost matches our method, highlighting the importance of the our feedforward pose estimation.

For gain on point accuracy, we attribute the gain to the photometric loss acting as an additional dense supervision signal, which also gets benefits from our robust pose estimation. Fig. H shows that our representation has much lower errors on planar road areas comparing to the model with per-pixel representation. With an accurate pose, the loss on the rasterized image effectively encourages

Table G: **Pose estimation with PnP+RANSAC.** Our method with PnP+RANSAC also outperforms direct competitors, validating the higher accuracy of our estimated geometry than others. Also our direct feedforward estimator achieves ∼16.6% higher accuracy than this baseline, further demonstrating its robustness over noise-sensitive iterative solvers. ATE and RPE translation (RPE$_{trans}$) and rotation (RPE$_{rot}$) are reported.

| Method | Stereo4D (Jin et al., 2025) | | | Bonn (Palazzolo et al., 2019) | | | Sintel (Butler et al., 2012) | | |
|---|---|---|---|---|---|---|---|---|---|
| | ATE | RPE$_{trans}$ | RPE$_{rot}$ | ATE | RPE$_{trans}$ | RPE$_{rot}$ | ATE | RPE$_{trans}$ | RPE$_{rot}$ |
| MonST3R (Zhang et al., 2025a) | 0.0458 | 0.0647 | 0.805 | 0.0031 | 0.0043 | 0.544 | 0.0290 | 0.0410 | 1.080 |
| St4RTrack (Feng et al., 2025) | 0.0602 | 0.0851 | 0.729 | 0.0024 | 0.0034 | 0.400 | 0.0231 | 0.0326 | 0.884 |
| UFO-4D (Ours, with PnP+RANSAC) | 0.0120 | 0.0170 | 0.205 | 0.0024 | 0.0033 | 0.293 | 0.0168 | 0.0238 | 0.332 |
| **UFO-4D (Ours, with the pose head)** | **0.0101** | **0.0142** | **0.179** | **0.0020** | **0.0028** | **0.271** | **0.0122** | **0.0172** | **0.298** |

the model to respect multi-view stereo constraints, guiding the estimated point to be geometrically consistent with the parallax in both views. This allows the model to better leverage both images as geometric supervision, whereas baselines only have view-independent sparse per-point supervision.

## A.10 DISCUSSION ON THE BETTER POSE ACCURACY

Table 3 in the main paper demonstrates that our method achieves the best pose accuracy among direct competitors. We further investigate the source of the gain to determine whether it originates from our feedforward estimation or the improved geometry. To answer the question, we also estimate relative pose via PnP+RANSAC from our predicted Gaussian centers.

The answer is both. As in Table G, our method with PnP with RANSAC still substantially outperforms other methods. This validates the higher accuracy of our estimated geometry than others, indicating that estimated points are aligned well with direction of each camera ray. Furthermore, our direct feedforward estimator achieves ∼16.6% higher accuracy than this PnP baseline, further demonstrating its robustness over noise-sensitive iterative solvers.

**The usage of Large Language Models (LLMs) in paper writing.** The paper got helps from LLMs for polishing writing and grammar correction.

