# OpenReview forum: "UFO-4D: Unposed Feedforward 4D Reconstruction from Two Images"
_ICLR.cc/2026/Conference — ICLR 2026 Poster_

### Official Review · Reviewer_aizK · 2025-10-30

**Soundness:** 3
**Presentation:** 3
**Contribution:** 3
**Rating:** 4
**Confidence:** 4

**Summary:**

The paper introduces UFO-4D, a feedforward model that takes two unposed images (with intrinsics) and predicts a dynamic 3D Gaussian Splat representation in a canonical frame plus the relative camera pose. A single differentiable 3DGS rasterizer is used to render RGB, pointmaps, and scene flow at arbitrary times and viewpoints by swapping rendered attributes through the same rasterizer. Training mixes supervised losses with photometric and edge-aware smoothness losses on rendered outputs. Strong results are reported on Stereo4D, Bonn, and KITTI, including 4D spatio-temporal interpolation from just two frames.

**Strengths:**

1. Unified explicit 4D design. Casting image/geometry/flow supervision through one renderer on 3D Gaussians without any additional regression heads is elegant and practical, aligning with the efficiency of 3DGS splatting while extending it to 4D prediction.
2. Quantitative gains. Clear improvements vs Zero-MSF and DynaDUSt3R on multiple datasets, suggesting the rendering supervision is beneficial for motion estimation.
3. Good ablation studies on training data, including single-dataset and mixed-dataset configurations, clarifying domain-transfer trade-offs.

**Weaknesses:**

1. Pose metrics are missing. Pose is central but quantitative metrics aren’t reported.
2. Lack of evaluations. Additional comparisons would strengthen the paper, particularly to dynamic reconstruction settings like **MonST3R [1]**, methods that jointly estimate pointmap and scene flow such as **Dynamic Point Maps [2]**, **St4RTrack [3]**, **POMATO [4]**, and classic scene-flow estimators like **RAFT-3D [6]**.
3. As noted in the paper’s limitations, the method is designed and evaluated for two views; there is no demonstrated extension to multi-view or longer sequences. This is problematic for dynamic scenes where motion reasoning benefits strongly from multi-view temporal context. Compared with several works [2,3,4,5] that do scale to multi-view settings, restricting to only two views appears limiting for long-range tracking, occlusion handling, and temporal consistency.
4. Lack of large-baseline or large-motion evaluation. The examples and datasets used in the paper mostly appear to involve small baselines and moderate motion. It would strengthen the work to include results on large-baseline or high-motion scenarios, or to evaluate on datasets such as MPI-Sintel that feature stronger motion variation. Demonstrating robustness in these challenging regimes would better validate the method’s generalization and motion reasoning capabilities.
5. Figure issue (Fig. 7): In the last example, the image and depth outputs appear swapped.

[1] Zhang, Junyi, et al. "Monst3r: A simple approach for estimating geometry in the presence of motion." ICLR’24.

[2] Sucar, Edgar, et al. "Dynamic Point Maps: A Versatile Representation for Dynamic 3D Reconstruction." ICCV’25.

[3] Feng, Haiwen, et al. "St4rtrack: Simultaneous 4d reconstruction and tracking in the world." ICCV’25.

[4] Zhang, Songyan, et al. "POMATO: Marrying Pointmap Matching with Temporal Motion for Dynamic 3D Reconstruction." ICCV’25.

[5] Han, Jisang, et al. "D^ 2USt3R: Enhancing 3D Reconstruction with 4D Pointmaps for Dynamic Scenes." NeurIPS’25.

**Questions:**

1. The motion and depth maps in the qualitative figures appear blurrier than those from ZeroMSF. Is this softness an inherent effect of the Gaussian representation and splatting, or does it result from the rendering resolution or regularization losses used during training?
2. Does the proposed method still perform reliable 4D interpolation when the two input views have a large baseline or significant motion?

---

> ### Author Response · Authors · 2025-11-25
>
> We sincerely appreciate the reviewer's time and helpful feedback and suggestions. We are more than happy to answer any questions if there are any remainings.
>
> ---
>
> > **Pose metrics are missing. Pose is central but quantitative metrics aren’t reported.**
>
> Please kindly refer to our response in the common response (``Relative pose evaluation``). Our method substantially outperforms baselines (MonST3R and St4RTrack) on Stereo4D, Bonn, and Sintel, without any extra solver.
>
> ---
>
> > **Additional comparison with more baseline methods.**
>
> To further highlight the strength of the paper, we include additional baselines (MonST3R and St4RTrack) an Sintel for comparison.
>
> We include the comparison table in the common response (``Evaluation with more baseline methods and datasets``). Our method substantially outperforms all baselines for 3D motion and relative pose estimation and performs competitively for geometry estimation.
>
> Per reviewer guidelines, Dynamic Point Maps, St4RTrack, and POMATO (ICCV 2025) are concurrent work, published after the submission date. That said, we prioritized St4RTrack for comparison due to its high relevance, excluding Dynamic Point Maps (unavailable code) and POMATO (insufficient documentation preventing reproduction). However, as POMATO shares St4RTrack's formulation, we expect their comparable performance. We will update comparison with RAFT-3D in the revision.
>
> ---
>
> > **As noted in the paper’s limitations, the method is designed and evaluated for two views; there is no demonstrated extension to multi-view or longer sequences. This is problematic for dynamic scenes where motion reasoning benefits strongly from multi-view temporal context. Compared with several works [2,3,4,5] that do scale to multi-view settings, restricting to only two views appears limiting for long-range tracking, occlusion handling, and temporal consistency.**
>
> Thanks for the feedback.
> We specifically target the unposed two-frame setting to establish a feedforward baseline for dynamic 3D Gaussians.
> In this challenging under-constrained setup, we demonstrate joint 2D/3D estimation and 4D interpolation via differentiable rasterization.
> Similar to DUSt3R-derived works, such as MonST3R (dynamic scene), NoPoSplat (NVS static scene), VGGT (multi-view static scene), our work opens up the new capability in the feedforward setup.
>
> Regarding the concurrent works [2–5], we note that their multi-view settings largely rely on post-processing concatenated pairwise outputs rather than inherent feature-level temporal reasoning.
> However, we agree that multi-extension is a valuable objective.
> We believe our unified representation serves as a strong baseline to build upon using multi-view architectures (e.g., VGGT) or video generative models.
>
> ---
>
> > **Lack of large-baseline or large-motion evaluation. Results on large-baseline or high-motion scenarios. Evaluation on datasets such as MPI-Sintel that feature stronger motion variation. Demonstrating robustness in these challenging regimes.**
>
> Please refer to the common response (`Test on large baseline, motion, and camera pose`) and `visual_example.html` for our qualitative assessment on large baselines and high motion.
>
> On Sintel, our method performs competitively with state-of-the-art baselines (`Evaluation with more baseline methods and datasets`) while achieving substantially superior relative pose accuracy (`Relative pose evaluation`).
>
>
> ---
>
> > **Swapped image and depth in Figure issue**
>
> Thank you! They are corrected.
>
> ---
>
>
> > **The motion and depth maps in the qualitative figures appear blurrier than those from ZeroMSF. Is this softness an inherent effect of the Gaussian representation and splatting, or does it result from the rendering resolution or regularization losses used during training?**
>
> ZeroMSF's clear depth boundaries highly likely stem from its use of large-scale synthetic datasets (SHIFT, Dynamic Replica, MOVi-F, Spring) with dense annotation. Interestingly their motion is blurry due to reliance on RAFT pseudo-labels.
>
> Our softness primarily arises from the sparse annotations in Stereo4D (DynaDUSt3R (in Fig. 4) share the same trait) and our smoothness regularization.
> Splatting sometimes yields blurry motion boundaries during 4D interpolation.
> We plan to address this with improved regularization strategies in future work.
>
>
>
> ---
>
> > **Does the proposed method still perform reliable 4D interpolation when the two input views have a large baseline or significant motion?**
>
> As demonstrated in the common response `Test on large baseline, motion, and camera pose` and attached `visual_example.html`, our model robustly produces all estimates from *just two unposed images*, effectively handling large object and camera motions with superior qualitative results.
>
> While motion estimation may degrade in extreme cases (*eg*, minimal image overlap), the recovered camera pose and 3D geometry remain highly robust.

---

### Official Review · Reviewer_W48t · 2025-10-30

**Soundness:** 4
**Presentation:** 4
**Contribution:** 3
**Rating:** 8
**Confidence:** 4

**Summary:**

This paper proposes a method to predict appearance and geometry (in the form of gaussian splats, motion and cameras) from two views. This is done in a feedforward manner and the main novelty is in connecting 2D motion/geometry supervision (in the form of point maps and scene flow) to the 3D outputs of the network, which is done by reusing the gaussian splats rendering aparatus for non-RGB rendering.

**Strengths:**

Although there are several papers tackling the general problem of 4D estimation (many correctly mentioned in the related work), this contribution stands out as a simple idea that is very well executed. Namely, the idea of rasterizing geometric and motion information into 2D to supervise it with scene flow and point maps (i.e. LIDAR) from real datasets seems somewhat novel (if a little obvious in hindsight, like all good ideas). This differs from approaches that demand 3D supervision, which often limits them to synthetic data, and approaches based purely on photometric supervision, which are plagued with degenerate solutions in the case of motion data (due to the inherent ambiguity of camera vs object motion).

The paper is written very clearly and the experiments are extensive, with significant gains. These are slightly tempered by the pretraining being based on recent SOTA weights, but the method brings a clear advantage.

**Weaknesses:**

For being based on NoPoSplat and MASt3r, a direct comparison with those methods where appropriate is needed. This would quantify the actual gains the method brings. However, it is understood that broadening the applicability of the original moodels can be a sufficient justification for the proposal.

As a minor aside on clarity, the smoothness loss should be written explicitly to make the paper self-contained, since it is an important factor for performance.

**Questions:**

Mostly I would like to know that a comparison is underway or that preliminary experiments indicated the gains are worth it, on a apples-to-apples comparison. This would turn the rating into a strong accept.

As the contribution is predicated on a simple idea, if other reviewers bring up (published) papers that call its priority into question, I would like to see a proper distinction between the methods.

---

> ### Author Response · Authors · 2025-11-25
>
> We sincerely appreciate the reviewer's positive recognition of our work and constructive feedbacks.
>
> ---
>
> > **For being based on NoPoSplat and MASt3r, a direct comparison with those methods where appropriate is needed. This would quantify the actual gains the method brings. However, it is understood that broadening the applicability of the original moodels can be a sufficient justification for the proposal.**
>
> The baselines (NoPoSplat and MASt3R) are designed to work on static scenes. Especially NoPoSplat is tied to their output representation, static 3D Gaussians.
> Our method broadens the applicability to handle dynamic scenes for 3D point, motion, and camera pose estimation, and also 4D interpolation.
> Notably, our method substantially outperforms ZeroMSF, which is already known to surpass MASt3R in dynamic settings.
> However, we agree that a direct comparison is valuable; we will include the results in the revision.
>
>
>
> ---
>
> > **As a minor aside on clarity, the smoothness loss should be written explicitly to make the paper self-contained, since it is an important factor for performance.**
>
> Thank you for the suggestion. We include detail formulations in Eq. (7c).
>
> ---
>
> > **Mostly I would like to know that a comparison is underway or that preliminary experiments indicated the gains are worth it, on a apples-to-apples comparison. This would turn the rating into a strong accept.**
>
> Thank you for your suggestion. Please kindly refer to the common response (``Apples-to-apples architecture comparison``).
>
> When comparing baselines trained under our protocol, our method demonstrates strong performance gains, especially on KITTI, highlighting the benefits of our feedforward dynamic 3DGS representation and differentiable rasterizer.
>
> ---
>
> > **As the contribution is predicated on a simple idea, if other reviewers bring up (published) papers that call its priority into question, I would like to see a proper distinction between the methods.**
>
> MonST3R (a published work) is designed to output 3D point maps and lacks capability of 3D motion estimation.
> As detailed in our common response (*Evaluation with more baseline methods and datasets* and *pose estimation*), our method outperforms MonST3R in both geometry and pose accuracy.
>
> There are other unpublished, concurrent works, Dynamic Point Maps, St4RTrack, POMATO, which share strong similarity to each other. They simply append an extra head for 3D motion or 3D position at next time step.
> Unlike them, our method outputs a single, holistic representation (*ie*, dynamic 3D Gaussians) which allows to rasterize 2D and 3D signals but also enable high-fidelity 4D spatio-temporal interpolation.
> Our method substantially outperforms St4RTrack on all tasks (*ie*, geometry, motion, and pose).
>
>
> ```
> [1] MonST3R: A simple approach for estimating geometry in the presence of motion, ICLR 2024
> [2] Dynamic Point Maps: A Versatile Representation for Dynamic 3D Reconstruction, ICCV 2025.
> [3] St4rtrack: Simultaneous 4d reconstruction and tracking in the world, ICCV 2025.
> [4] POMATO: Marrying Pointmap Matching with Temporal Motion for Dynamic 3D Reconstruction, ICCV 2025.
> [5] D^2USt3R: Enhancing 3D Reconstruction with 4D Pointmaps for Dynamic Scenes, to appear at NeurIPS 2025.
> ```

---

> > ### Comment · Reviewer_W48t · 2025-11-27
> >
> > Thank you for the detailed answer. I appreciate the improvements and the new comparisons are convincing. I agree with the authors that other concurrent works brought up should not require any direct comparison, but mentioning them as concurrent works is nice.
> >
> > On the subject of outperforming St4rTrack, it's important to distinguish between the short-term dynamics tasks under study, and long-term tracking (which many point-tracking works focus on), which is not evaluated here -- this is OK, but the distinction of which areas each method is supposed to be stronger should be clear.

---

> > > ### Author Response · Authors · 2025-11-28
> > >
> > > Thank you for the encouraging feedback and the constructive suggestions, which have helped strengthen our work.
> > >
> > > We fully agree that St4RTrack focuses on long-term tracking, a distinct task where it has specific strengths.
> > > We have added a note in Sec. 4.1 to clarify this distinction.
> > >
> > > Thank you again, and please feel free to let us know if there are any further questions.

---

### Official Review · Reviewer_jDQa · 2025-11-01

**Soundness:** 3
**Presentation:** 3
**Contribution:** 3
**Rating:** 4
**Confidence:** 4

**Summary:**

This paper introduces UFO-4D, a unified feedforward architecture for reconstructing dense 4D scenes from two images without known camera poses. The method simultaneously predicts dynamic 3D Gaussian splats and the relative camera transformation in one forward pass, creating an explicit 4D scene representation that jointly recovers 3D structure, motion fields, and camera pose while enabling spatio-temporal interpolation. By employing differentiable rasterization, the framework integrates both self-supervised photometric constraints and supervised signals for geometry and motion, establishing a strong coupling between depth and motion prediction. Comprehensive evaluations show state-of-the-art results across multiple geometry and motion benchmarks, supported by qualitative and quantitative analysis, ablation studies, and novel 4D interpolation use cases.

**Strengths:**

- **Unified feedforward architecture:** Unlike previous works that obtain geometry and motion estimation with separate steps, the proposed UFO-4D offers an unified 4D representation capable of solving multiple downstream perception tasks jointly which has been found to be beneficial in other domains.
- **Leveraging 4DGS representation:** By leveraging 4D Gaussian Splatting as the representation, this allows the model to be trained with additional photometric supervision, and also allow other estimations such as motion vectors to be easily rasterized allowing additional self-supervised signals, and allowing all signals to train a joint representation.
- The overall architecture is simple and straightforward, with the details of the architecture being explicitly mentioned, making it easy for re-implementation of the method.

**Weaknesses:**

- **Lack of direct comparison:** Although I agree that leveraging a representation that tightly couples both pointmap and motion estimation will further boost performance, it is difficult to understand the benefits that come from adopting this new representation itself as the proposed method also utilizes a very recently open-sourced dataset Stereo4D for training. It would be nice to include a comparison of baseline method trained with the same dataset recipe to better highlight the advantages of the proposed representation.
- **Small baselines:** The qualitative figures shown in the paper seem to take two frames with small baselines as input. As a result, it is difficult to understand how well the method is estimation motions when given large baselines (two frames that are far away) as input.
- **Limited evaluation:** Adding more evaluation results in dynamic datasets such as Sintel[1] or TUM-Dynamics[2] could better highlight the strength of the paper. Also adding evaluation of the estimated camera pose for all datasets would be needed to emphasize the importance of adopting the proposed representation.

### References

---

[1] Butler, Daniel J., et al. "A naturalistic open source movie for optical flow evaluation." European conference on computer vision. Berlin, Heidelberg: Springer Berlin Heidelberg, 2012.

[2] Sturm, Jürgen, et al. "A benchmark for the evaluation of RGB-D SLAM systems." *2012 IEEE/RSJ international conference on intelligent robots and systems*. IEEE, 2012.

**Questions:**

Q1. How does UFO4D estimate relative camera pose? How does the performance of camera pose estimation compare with prior works?

Q2. As mentioned in the discussion of the Stereo4D dataset, there seem to be pixels in both images where the tracking and correspondence information is not provided in the dataset. For these pixels, what does the velocity vector in the predicted Gaussian learn? In addition, how sparse are these signals during training? Can other supervision from dense losses compensate for this sparsity?

Q3. How robust is UFO4D to large baseline images?

Q4. Why does UFO4D initialize the weights from Mast3r instead of learning directly from the finetuned version MonST3R? It seems that training could be further boosted and become more efficient when initialized from MonST3R.

With several questions and unclear parts remaining, I would like to recommend border reject as my initial rating. However, with my questions being resolved, I am eager to raise my score.

---

> ### Author Response · Authors · 2025-11-25
>
> We sincerely appreciate the reviewer's time and helpful feedback and suggestions. We are more than happy to answer any questions if there are any remainings.
>
> ---
>
> > **A comparison of baseline method trained with the same dataset recipe to better highlight the advantages of the proposed representation**
>
> Thank you for the suggestion. Please refer to the common response (`Apples-to-apples architecture comparison`).
>
> We train baseline-equivalent models with the same training recipe (*e.g.,* dataset, augmentation, hyperparameters). We observe strong performance gain especially on KITTI, benefiting from our feedforward dynamic 3DGS representation with differentiable rasterizer.
>
> ---
>
> > **Test on large-baseline input (two frames that are far away)** and  **Q3. How robust is UFO4D to large baseline images?**
>
> Please refer to our answer in the common response: ``Test on large baseline, motion, and camera pose``.
>
> Our model robustly estimates dynamic 3D Gaussians from *just two unposed images*, even in the presence of large object and camera motion. While motion may degrade in extreme cases, *eg* minimal image overlap, the recovered camera pose and 3D geometry remain highly robust.
>
> ---
>
> > **Adding more evaluation results in dynamic datasets**
>
> To further highlight the strength of the paper, we include additional baselines (MonST3R and St4RTrack) for comparison and add Sintel for evaluation.
>
> We include the comparison and discussion in the common response (``Evaluation with more baseline methods and datasets``). Our method substantially outperforms all baselines for 3D motion estimation and performs competitively for geometry estimation.
>
>
> ---
>
> > **Q1. How does UFO-4D estimate relative camera pose? How does the performance of camera pose estimation compare with prior works?**
>
> We include the relative pose evaluation in the common response (``Relative pose evaluation``). Our method substantially outperforms baseline methods (MonST3R and St4RTrack) on Stereo4D, Bonn, and Sintel.
>
>
> ---
>
> > **Q2. As mentioned in the discussion of the Stereo4D dataset, there seem to be pixels in both images where the tracking and correspondence information is not provided in the dataset. For these pixels, what does the velocity vector in the predicted Gaussian learn? In addition, how sparse are these signals during training? Can other supervision from dense losses compensate for this sparsity?**
>
> In our experiments, valid ground truth covers only ~10.3% of pixels in Stereo4D. Our method effectively learns velocity in the remaining regions through its explicit Dynamic 3D Gaussian representation with differentiable rasterizer. Unlike per-pixel methods (*e.g.*, St4RTrack, DynaDUSt3R), a single 3D Gaussian contributes to multiple pixels, which allows gradients from sparse signals to propagate to the underlying primitives via our differentiable rasterizer.
>
> Furthermore, we leverage a dense photometric loss to encourage constancy between the rendered images and input images. This provides indirect supervision for all pixels including unannotated regions, forcing the explicit 4D primitives to satisfy both sparse 3D annotations and dense 2D image evidence. Ablation (Table 4, main paper) confirms that adding this loss significantly improves motion EPE (from 0.072 to 0.064). Fig. 6 (main paper) qualitatively demonstrates its importance for sharp motion and depth boundaries (No image $\rightarrow$ Full model).
>
>
> ---
>
> > **Q4. Why does UFO4D initialize the weights from Mast3r instead of learning directly from the finetuned version MonST3R? It seems that training could be further boosted and become more efficient when initialized from MonST3R.**
>
> Thank you for the suggestion. The table below compares the two initializations on the Stereo4D, KITTI, and Bonn datasets while using the training setup from the architecture comparison study (*ie*, $256\times256$ training resolution, 60k iteration steps).
>
> Interestingly MASt3R initialization yields overall better accuracy. MonST3R provides better 3D point accuracy on Bonn and KITTI. We presume that the finetuned checkpoint of MonST3R is particularly optimized for Bonn, as observed their competitive accuracy on Bonn in Table 1 in the main paper (benchmark table), although our model does not necessarily need this specific prior.
>
>
> | | | | Stereo4D | |  | |  KITTI |  |  | Bonn |  |
> | :---- | :---: | :---: | :---: | :---: | :---: | :---: | :---: | :---: | :---: | :---: | :---: |
> |  | | PSNR | Point EPE | Motion EPE |  | PSNR | Point EPE | Motion EPE |  | PSNR | Point EPE |
> | Init. from MASt3R | | **24.52** | **0.781** | **0.058** |  | **19.61** | 3.553 | **0.244** |  | 26.61 | 0.211 |
> | Init. from MonST3R | | 24.20 | 0.868 | 0.059 |  | 19.51 | **3.260** | 0.258 |  | **26.55** | **0.155** |

---

> > ### Comment · Reviewer_jDQa · 2025-11-25
> >
> > I highly appreciate the author's work and the volume of experiments conducted for the rebuttal. The results are interesting, and I am more eager to raise my score. Could the authors answer some of my follow-up questions?
> >
> > > **Q1.** The performance of relative pose estimation is interesting.
> > ---
> > > **Q1-1.** Does this also increase the accuracy of the estimated geometry, similarly increasing the performance when relative pose is estimated with a similar PnP + RANSAC by leveraging the mean positions of the estimated Gaussians as 3D points?
> >
> > > **Q1-2.** How does the pose token learn to estimate an accurate relative camera pose in dynamic scenes? In particular, to estimate an accurate camera pose in dynamic scenes, it is important to only focus on the static regions. Are there any observations of the pose token attending to these static regions in the attention?
> >
> > > **Q1-3.** For Q1-1, if the pose accuracy drops for the case of PnP + RANSAC, would adding another self-supervised signal to match the performance of camera pose estimation from the pose tokens and the estimation of 3D geometry boost the overall performance?
> > ---
> > > **Q2.** Thank you for the experiment of training different architectures with the same setup. I highly appreciate the work dedicated to this experiment regarding the short rebuttal period. A simple question would be, why does the proposed architecture boost the performance for the KITTI dataset? In my experience, the KITTI dataset does not include a lot of motions, which makes the results interesting.

---

> > > ### Author Response · Authors · 2025-11-28
> > >
> > > Thank you very much for the follow-up, interesting, and insightful questions! If there are any remainings, we are more than happy to answer any questions.
> > >
> > > > **Q1-1. Does this also increase the accuracy of the estimated geometry, similarly increasing the performance when relative pose is estimated with a similar PnP + RANSAC by leveraging the mean positions of the estimated Gaussians as 3D points?**
> > >
> > > As in the table below, estimating pose via PnP+RANSAC from our Gaussian centers still substantially outperforms other methods.
> > > This validates the higher accuracy of our estimated geometry than others.
> > > Furthermore, our direct feedforward estimator achieves ~16.6% higher accuracy than this PnP baseline, which also demonstrates its robustness over noise-sensitive iterative solvers.
> > >
> > >
> > >
> > > | Pose estimation |  ||  Stereo4D |  |  | | Bonn |  |  | | Sintel Final |  |
> > > | :---- | :---- | :---: | :---: | :---: | :---: | :---: | :---: | :---: | :---: | :---: | :---: | :---: |
> > > |  |  | ATE | RPE\_t | RPE\_r |  | ATE | RPE\_t | RPE\_r |  | ATE | RPE\_t | RPE\_r |
> > > | MonST3R |  | 0.0458 | 0.0647 | 0.805 |  | 0.0031 | 0.0043 | 0.544 |  | 0.0290 | 0.0410 | 1.080 |
> > > | St4RTrack |  | 0.0602 | 0.0851 | 0.729 |  | 0.0024 | 0.0034 | 0.400 |  | 0.0231 | 0.0326 | 0.884 |
> > > | **Ours (PnP+RANSAC)** |  | 0.0120 | 0.0170 | 0.205 |  | 0.0024 | 0.0033 | 0.293 |  | 0.0168 | 0.0238 | 0.332 |
> > > | **Ours** |  | **0.0101** | **0.0142** | **0.179** |  | **0.0020** | **0.0028** | **0.271** |  | **0.0122** | **0.0172** | **0.298** |
> > >
> > >
> > >
> > >
> > > > **Q1-2. How does the pose token learn to estimate an accurate relative camera pose in dynamic scenes? In particular, to estimate an accurate camera pose in dynamic scenes, it is important to only focus on the static regions. Are there any observations of the pose token attending to these static regions in the attention?**
> > >
> > > Through supervised learning, our model learns to predict accurate relative pose by isolating static regions.
> > > To confirm this, we visualize cross-attention scores between the camera pose token and image tokens in Appendix Fig. G.
> > > As a reference, we also visualize a static baseline for each scene by using two identical images as input.
> > > We observe that cross-attention maps at specific blocks (*eg,* 8th, 11th, and 12th) exhibit higher attention scores in static regions, effectively suppressing attention on moving objects.
> > >
> > >
> > > > **Q1-3. For Q1-1, if the pose accuracy drops for the case of PnP + RANSAC, would adding another self-supervised signal to match the performance of camera pose estimation from the pose tokens and the estimation of 3D geometry boost the overall performance?**
> > >
> > > Given that the pose accuracy drops with PnP, we could conceptually use the self-supervised loss for consistency between the direct pose output and the geometry-drived pose (*ie* via PnP) to improve performance, expecially on geometry.
> > >
> > > However, our model already enforces this consistency implicitly through differentiable 4D rasterization.
> > > Because the predicted pose is explicitly used for rendering outputs at the target view, any mismatch between the estimated geometry and pose is directly penalized by losses on rendered image, motion, and geometry.
> > > Thus the new self-supervised loss may not give extra gain.
> > >
> > > That said, we agree that it is an interesting empirical question if the explicit consistency loss further improves the correctness of the output.
> > > It can be especially beneficial for mitigating effects from noisy annotations in training datasets (*eg*, Stereo4D).

---

> > > > ### Author Response · Authors · 2025-11-28
> > > >
> > > > > **Q2. Thank you for the experiment of training different architectures with the same setup. I highly appreciate the work dedicated to this experiment regarding the short rebuttal period. A simple question would be, why does the proposed architecture boost the performance for the KITTI dataset? In my experience, the KITTI dataset does not include a lot of motions, which makes the results interesting.**
> > > >
> > > > We find that the huge accuracy gain on KITTI is from both the accurate pose estimation and better supervision with our photometric losses.
> > > >
> > > > Our accuracy gain for motion on KITTI primarily originates from our accurate pose estimation.
> > > > Since KITTI evaluates motion in local camera coordinates, predictions in the canonical camera coordinate need to be transformed using estimated poses.
> > > > Also KITTI includes many scenes with large ego-motion moving forward. Thus, accurate pose is critical for achieving better motion accuracy.
> > > > For baselines, we use PnP-RANSAC that follows their formulation.
> > > > In contrast, our method utilizes a more accurate pose from our feedforward estimation, leading better motion accuracy in the end.
> > > >
> > > > We validated this source of improvement by applying the pose head to a per-pixel baseline as well (*ie* equivalent to DynaDUSt3R with the pose head.).
> > > > As in the table below, its motion accuracy (0.245) on KITTI is now very similar to our method (0.244), highlighting the importance of the our feedforward pose estimation.
> > > > (Interestingly accuracy on other datasets degrade a bit possibly due to sharing the network capacity for pose estimation.)
> > > >
> > > > | Output representation | Pose estimator |  | Stereo4D |  |  | KITTI |  |  | Bonn |
> > > > | :---- | ----- | :---- | :---: | ----- | :---- | :---: | ----- | :---- | :---: |
> > > > |  |  |  | Motion EPE | Point EPE |  | Motion EPE | Point EPE |  | Point EPE |
> > > > | **Ours** | Pose head |  | **0.057** | **0.781** |  | **0.244** | **3.553** |  | 0.211 |
> > > > | per-pixel point and motion | Pose head |  | 0.060 | 0.811 |  | 0.245 | 4.396 |  | 0.206 |
> > > > | per-pixel point and  motion | PnP RANSAC |  | 0.058 | 0.809 |  | 0.283 | 4.567 |  | **0.196** |
> > > >
> > > > For point estimation, we attribute its gain to the photometric loss which acts as an additional dense supervision signal, enabled by our robust pose estimation.
> > > > Figure H in Appendix shows that our representation has much lower errors on planar road areas comparing to the model with per-pixel representation.
> > > > With an accurate pose, the loss on the rasterized image effectively encourages the model to respect multi-view stereo constraints, guiding the estimated point to be geometrically consistent with the parallax in both views.
> > > > This allows the model to better leverage both images as geometric supervision, whereas baselines only have view-independent sparse per-point supervision.

---

> > > > > ### Comment · Reviewer_jDQa · 2025-11-28
> > > > >
> > > > > Thank you for the authors for dedicating their time and effort to provide responses to my follow-up questions. I think the results are really interesting and both the additional experiments for Q1-1 and Q1-2 should be included in the paper as it reveals the contributions of the proposed framework better. After discussion, I think this paper has sufficient contributions to the field to be accepted, and I am raising my score to 6.
> > > > >
> > > > > * As OpenReview is not allowing me to raise my score, I will raise my score when the problem is fixed. Do not hesitate to notify me if I forget to modify my final score.

---

> > > > > > ### Author Response · Authors · 2025-12-03
> > > > > >
> > > > > > Thank you for your response and positive evaluation!
> > > > > >
> > > > > > Following the suggestion, we have included the additional analyses in Appendix A.8, A.9, and A.10.
> > > > > > We sincerely appreciate your time and the insightful questions that led the interesting analyses and findings.

---

### Official Review · Reviewer_Y37w · 2025-11-03

**Soundness:** 2
**Presentation:** 3
**Contribution:** 3
**Rating:** 4
**Confidence:** 4

**Summary:**

This paper focuses on a dynamic Gaussian reconstruction framework that infers 4D scenes from two images. The reconstructed 4D scenes enables motion extraction and novel view synthesis. The experiments demonstrate the effectiveness of the proposed method.

**Strengths:**

1. The paper focuses on a cutting-edge task that directly predicts Gaussian attributes from unposed images. The task is applicable of many downstream applications.
2. This paper implement a standard and simple network with pure ViTs, which can be potentially scaled up to large datasets and parameters.

**Weaknesses:**

1. The results are not convincing. This paper does not show enough visualization on the rendering quality of the reconstructed 3DGS at different novel views. And no videos are provided as the supplementary for a clear visualization on the reconstruction quality. So I am not sure whether the generated 3DGS are of high-quality. Given the current figures, I do not see significant performance gain compared to only predict colored points, which weakens the insight of predicting Gaussian primitives.
2. A video showing novel view synthesis results are required.
3. I found that the input image pairs are often of very similar viewpoints. I wonder if the method can handle pairs of images that contains large offsets in camera poses?
4. A comparison with 4DGT is recommended since 4DGT also focuses on predicting dynamic Gaussians. Since 4DGT requires camera poses as input, you can simply apply an off-the-shelf camera pose estimation model first.

**Questions:**

Please refer to the weaknesses above.

---

> ### Author Response · Authors · 2025-11-25
>
> We sincerely thank the reviewer's time and constructive feedback and suggestions. If there are any remaining questions, we are more than happy to answer them.
>
> ---
>
> > **3DGS visualization and video results**
>
> The attached **``visual_example.html``** demonstrates our high-quality 4D interpolation results, where we rasterize image, depth, and motion at interpolated novel views from just two unposed images.
> Even in scenarios with significant object and camera motion, our model yields robust predictions.
> This highlights the critical advantage of predicting Gaussian primitives: unlike per-pixel point estimation methods (*eg*, ZeroMSF, St4RTrack, DynaDUSt3R) which suffer from depth ordering issues during re-projection, our explicit 3DGS representation naturally handles occlusions.
>
>
> ---
>
> > **"I wonder if the method can handle pairs of images that contains large offsets in camera poses?"**
>
> Please refer to the answer in the common response: ``Test on large baseline, motion, and camera pose``.
>
> Our model robustly outputs all estimates from *just two unposed images*, even in the presence of large object and camera motion. While motion estimation may degrade in extreme cases, *eg* minimal image overlap, the recovered camera pose and 3D geometry remain highly robust.
>
>
> ---
>
> > **Comparison with 4DGT**
>
> According to the reviewer guideline of ICLR 2026, 4DGT is a concurrent, unpublished work.
>
> ```
> ... "if a paper was published (i.e., at a peer-reviewed venue) on or after July 24, 2025, authors are not required to compare their own work to that paper. Note that arXiv is not considered a peer-reviewed venue." ...
> ```
>
> That said, we plan to compare with 4DGT and update results soon. Due to time constraints and the need to allocate compute resources efficiently, we have been focusing on the essential ablation studies and comparison.

---

### Author Response · Authors · 2025-11-25
**Common response**

We sincerely thank all the reviewers for the valuable time and effort they dedicated to assessing our submission. The constructive feedback and insightful suggestions have been extremely helpful. We have organized our response below by first addressing the  common questions raised by multiple reviewers, and then providing detailed, individual replies to each reviewer's specific comments.

We have also updated our main paper with Appendix and attached supplementary materials for visual results. Please refer to **``visual_example.html``** for more qualitative results.

We appreciate this opportunity to revise our work based on your constructive comments and would be delighted to provide any additional clarification if any questions persist.

---

> **Test on large baseline, motion, and camera pose**

We include visual examples in the supplementary material (**``visual_example.html``**) for providing results on 4D interpolation and qualitative comparison.

**Qualitative comparison.**
Figure 4 in the main paper has been updated. Initially, *the first image was accidentally duplicated as the input image pair*; as corrected in the current version, the two input images have substantial view changes.


Our method outperforms competing methods by estimating more accurate depth and motion, even under challenging conditions such as large camera rotation and object motion. Ours effectively disentangles dynamic object motion from camera ego-motion, preserving sharp boundaries and geometric consistency.

**4D interpolation.** Our model successfully interpolates images, depth, and motion by rendering high-quality outputs from predicted dynamic 3D Gaussians. Even on scenarios with large object motion and camera motion, our model robustly predicts those estimates from *just unposed two images*.

In very challenging cases (*eg,* extremely large motion or minimal image overlaps), our model struggles with accurate motion estimation. However, camera pose and 3D geometry estimation remains robust.





> **Relative pose evaluation**
>
The table below summarizes pose accuracy on Stereo4D test split, Bonn, and Sintel Final. Given two-frame inputs, we estimate relative pose and report accuracy using standard metrics, ATE and RPE (translation and rotation).
Our method achieves the best performance across all datasets. Unlike MonST3R and St4RTrack, which obtain pose via a PnP solver with RANSAC, our approach is fully feed-forward. We demonstrate that direct estimation eliminates the need for time-consuming post-processing while simultaneously achieving higher accuracy.
Other baselines (*e.g.,* ZeroMSF or DynaDUSt3R) do not aim to output camera pose.

| Pose estimation |  ||  Stereo4D |  |  | | Bonn |  |  | | Sintel Final |  |
| :---- | :---- | :---: | :---: | :---: | :---: | :---: | :---: | :---: | :---: | :---: | :---: | :---: |
|  |  | ATE | RPE\_t | RPE\_r |  | ATE | RPE\_t | RPE\_r |  | ATE | RPE\_t | RPE\_r |
| MonST4R |  | 0.0458 | 0.0647 | 0.805 |  | 0.0031 | 0.0043 | 0.544 |  | 0.0290 | 0.0410 | 1.080 |
| St4RTrack |  | 0.0602 | 0.0851 | 0.729 |  | 0.0024 | 0.0034 | 0.400 |  | 0.0231 | 0.0326 | 0.884 |
| **Ours** |  | **0.0101** | **0.0142** | **0.179** |  | **0.0020** | **0.0028** | **0.271** |  | **0.0122** | **0.0172** | **0.298** |

---

> ### Author Response · Authors · 2025-11-25
> **Common response**
>
> > **Evaluation with more baseline methods and datasets**
>
> As suggested, we include additional baselines (MonST3R and St4RTrack) for comparison and add Sintel for evaluation.
>
>
> Our method outperforms all baseline methods by a large margin on 3D estimation.
> Though ZeroMSF is trained on a very large-scale dataset for driving scenes, EPE of our method is 3 times lower than that of ZeroMSF.
>
> | 3D motion estimation |  | Stereo4D |  (forward) |  | Stereo4D |  (backward) |  | KITTI |  |
> | :---- | :---: | :---: | :---: | :---: | :---: | :---: | :---: | :---: | :---: |
> |  |  | EPE | $\delta^{0.05}_\text{3D}$ |  | EPE | $\delta^{0.05}_\text{3D}$ |  | EPE | $\delta^{0.05}_\text{3D}$ |
> | DynaDUSt3R |  | 0.175 | 51.46 |  | 0.161 | 52.13 |  | 0.463 | 1.090 |
> | ZeroMSF |  | 0.164 | 45.63 |  | \- | \- |  | 0.442 | 16.20 |
> | St4RTrack |  | 0.373 | 26.62 |  | \- | \- |  | 0.751 | 4.01 |
> | **Ours** |  | **0.049** | **83.06** |  | **0.050** | **82.67** |  | **0.137** | **47.80** |
>
>
> Our method substantially outperforms all 4D estimation methods (DynaDUSt3R, ZeroMSF, and St4RTrack) for 3D point estimation and depth estimation.
> MonST3R and ZeroMSF performs comparable with our method on Sintel due to the usage of the Spring dataset for training (*ref.* Table 5 in the MonST3R paper: adding Spring improves accuracy on Sintel due to their closer domain).
>
>
> | Geometry estimation |  | | Stereo4D |  |  | | Bonn  |  |  | | KITTI  |  |  | | Sintel |  |
> | :---- | :---- | :---: | :---: | :---: | ----- | :---: | :---: | :---: | ----- | :---: | :---: | :---: | ----- | :---: | :---: | :---: |
> |  |  | EPE | Abs | $\delta{\lt} 1.25$ |  | EPE | Abs | $\delta{\lt} 1.25$ |  | EPE | Abs | $\delta{\lt} 1.25$ |  | EPE | Abs | $\delta{\lt} 1.25$ |
> | MonST3R |  | 1.504 | 0.191 | 74.75 |  | 0.188 | **0.062** | **96.22** |  | 2.797 | 0.107 | 87.61 |  | **3.607** | 0.341 | 57.61 |
> | DynaDUSt3R |  | 0.811 | 0.112 | 86.07 |  | 0.424 | 0.084 | 92.75 |  | 4.889 | 0.116 | 83.39 |  | 4.362 | **0.301** | 61.04 |
> | ZeroMSF |  | 1.662 | 0.220 | 76.18 |  | 0.208 | 0.079 | 91.58 |  | 3.392 | 0.107 | **92.28** |  | 3.637 | 0.333 | **63.86** |
> | St4RTrack |  | 1.467 | 0.209 | 71.26 |  | 0.181 | 0.067 | 95.17 |  | 4.731 | 0.125 | 82.86 |  | 4.145 | 0.357 | 57.14 |
> | **Ours** |  | **0.659** | **0.106** | **88.38** |  | **0.162** | 0.064 | 96.04 |  | **2.568** | **0.090** | 88.92 |  | 3.866 | 0.319 | 61.94 |
>
>
>
>
> ---
>
> > **Apples-to-apples architecture comparison**
>
> Thanks for suggesting this experiment.
>
> In order to decouple architectural benefits from implementation details, we trained baselines under the exact same protocol (our full-training protocol but with $256 \times 256$ resolution and 60k iterations) with output representations matching prior works:
>  - Per-pixel point and motion (equivalent to DynaDUSt3R and ZeroMSF).
>  - Per-pixel point only (equivalent to MonST3R ).
>
>
> | Architecture | Equivalent or closest models |  | Stereo4D |  |  | KITTI |  |  | Bonn |
> | :---- | :---- | :---- | :---- | :---- | :---- | :---- | :---- | :---- | :---- |
> |  |  |  | Motion EPE | Point EPE |  | Motion EPE | Point EPE |  | Point EPE |
> | Ours | **UFO-4D (Ours)** |  | 0.058 | **0.781** |  | **0.244** | **3.553** |  | 0.211 |
> | per-pixel point motion | DynaDUSt3R, ZeroMSF |  | **0.057** | 0.809 |  | 0.283 | 4.567 |  | **0.196** |
> | per-pixel point only | MonST3R |  | \- | 0.804 |  | \- | 4.562 |  | 0.201 |
>
>
> Our Dynamic 3DGS representation outperforms per-pixel baselines on Stereo4D and KITTI, demonstrating its effectiveness for dynamic scenes. Futher, our representation enables new capability of 4D interpolation.
>
> Our method is slightly behind on Bonn. We attribute this to Bonn's textureless regions (*e.g.,* walls), where our model predicts large-scale, overlapping Gaussians (Figure E, p21, Appendix). While this heavy overlap ensures surface continuity, it makes per-pixel point dependent on the weighted combination of multiple nearby Gaussians, making it prone to accumulated errors when there exists Gaussians with inaccurate 3D position nearby (Figure F, p21, Appendix).
> We quantitatively confirm this hypothesis in Table E in Appendix that depth accuracy degrades as the scale of the estimated 3D Gaussians increases.

---

### Meta-Review · Area_Chair_NyKV · 2026-01-07

**Summary:**

UFO-4D proposes a feed-forward method to reconstruct a dense 4D scene representation from two unposed images by predicting dynamic 3D Gaussians (geometry/appearance + 3D motion) and relative camera pose in one pass. A key design choice is reusing a single differentiable 3DGS rasterizer to render RGB, geometry, and motion signals, enabling both photometric self-supervision and coupled geometry/motion losses, and supporting 4D interpolation.

Before rebuttal, reviews were split: one strong accept highlighted the simplicity and execution quality, while three borderline reviews questioned (i) robustness/qualitative evidence beyond small-baseline cases, (ii) the lack of quantitative pose evaluation despite pose being central, and (iii) whether gains were attributable to the proposed representation/training design rather than training recipe/data.

The rebuttal and discussion substantially strengthen the evidence with additional qualitative material, pose metrics, broader comparisons, and controlled experiments isolating architectural effects.

**Reviewer Concerns:**

Addressed by rebuttal/discussion:

1. Pose metrics missing (aizK, jDQa): Authors added quantitative relative pose results (ATE/RPE) on Stereo4D, Bonn, and Sintel Final, and follow-up analysis comparing direct pose prediction vs PnP+RANSAC using Gaussian centers; they also provide attention visualizations supporting focus on static regions.

2. Large baseline / large motion + qualitative evidence (Y37w, jDQa, aizK): Authors provided qualitative examples via visual_example.html and discussion of behavior under large motion/pose changes; they acknowledge motion can degrade in extreme low-overlap cases.

3. More baselines/datasets (aizK, jDQa): Authors added MonST3R and St4RTrack comparisons and added Sintel evaluation.

4. Representation vs recipe confound; apples-to-apples (jDQa, W48t): Authors trained baseline-style per-pixel representations under the same protocol and reported comparative results to isolate the benefit of the dynamic 3DGS representation + rasterizer-based supervision.

Partially addressed / still outstanding (not correctness blockers):

1. The method remains focused on the two-frame setting; no multi-view/long-sequence temporal reasoning is demonstrated. This should be framed as a deliberate scope choice and not over-claimed as long-term tracking.

2. Some additional comparisons requested (e.g., classic scene-flow baselines such as RAFT-3D) are not included in the rebuttal record; authors indicate intent to add some comparisons in revision. This would strengthen the camera-ready but is not essential for acceptance given the expanded evaluation already provided.

3. The authors note some boundary “softness” and extreme-case motion failure modes; these should remain clearly stated with representative examples.

**Reviewer Scores:**

W48t: 8 → 8 (explicitly confirmed 8; rebuttal addressed the requested comparisons/clarifications).

jDQa: 4 → 6 (explicitly stated the paper should be accepted and intended to raise to 6).

aizK: 4 → 4 (some points addressed, but the two-view vs longer/multi-view scope concern remains; no follow-up indicating a score change).

Y37w: 4 → 4 (authors provided more qualitative material and discussed large motion, but no confirmation that the reviewer’s core doubts/video-related requests were resolved).

---

### Decision · Program_Chairs · 2026-01-26

Accept (Poster)